# Syndecans and glycosaminoglycans influence B-cell development and activation

Craig I McKenzie [1,2✉], Alexandra R Dvorscek[1,3], Zhoujie Ding [1], Marcus J Robinson [1], Kristy O'Donnell [1], Catherine Pitt [1], Daniel T Ferguson[4], Jesse Mulder [1], Marco J Herold [5,6,7,8], David M Tarlinton [1] & Isaak Quast [1✉]

## Abstract

Syndecans (SDCs) are glycosaminoglycan-containing cell surface proteins with diverse functions in the immune system with SDC1 (CD138) and SDC4 expressed in B-lineage cells. Here, we show that stem cells lacking either molecule generate fewer B-cell progenitors but give rise to mature B cells in vivo. Deletion of the plasma cell "marker" CD138 has no effect on homeostatic or antigen-induced plasma cell formation. Naive B cells express high SDC4 and encounter with cognate antigen results in transient CD138 upregulation and SDC4 loss, both further modulated by IL-4, IL-21, and CD40 ligation. SDC4 is downregulated on germinal center B cells and absent on most memory B cells. Glycosaminoglycans such as those attached to SDCs, and heparin, a commonly used therapeutic, regulate survival and activation of naive B cells by limiting responsiveness to cognate antigen. Conversely, ablation of SDC4 results in increased baseline and antigen-induced B-cell activation. Collectively, our data reveal B-cell activation- and subset-dependent SDC expression and show that SDC4 and GAGs can limit antigen-induced activation to promote B-cell survival and expansion.

**Keywords** B Cells; CD138; Syndecan-4; Heparin; Glycosaminoglycans
**Subject Categories** Immunology; Membranes & Trafficking; Signal Transduction

## Introduction

Syndecans (SDCs) are a multi-functional family of cell surface proteoglycans involved in interactions with the extracellular matrix, integrin clustering, and binding of growth factors to promote survival (Yang et al, 2022). These functions are in part mediated by long, unbranched chains of sulfated carbohydrates termed glycosaminoglycans (GAG) that are post-translationally added to SDCs and mediate the binding of positively charged residues, such as those within cytokines and growth factors (Song et al, 2021). Two major forms of GAGs are attached to SDCs: heparan sulfate (HS) and chondroitin sulfate (CS) (Gopal, 2020). In the context of B-cell biology, HS has been shown to influence cytokine signaling and integrin binding to B lymphocytes (Chen et al, 2023; Moreaux et al, 2009) and is predicted to interact with key signaling molecules such as the B-cell receptor (BCR) (Gomez Toledo et al, 2021; Levy et al, 1981). B cells express two members of the SDC family, HS-containing SDC1 (CD138) and SDC4 (SDC4), at differing stages of development (Reijmers et al, 2013) but the roles these SDCs play in B-cell maturation, activation, and differentiation remain poorly understood.

High CD138 is a canonical marker for plasma cells (PC) in mice and humans (Ellyard et al, 2004; Halliley et al, 2015; Hayashi et al, 1987; Lalor et al, 1992), but it is first expressed on B-cell precursors in the bone marrow (BM) (Sanderson et al, 1989). In mature B-cell subsets, low to intermediate CD138 expression is observed in a small population of follicular B cells (Lee et al, 2013). This has been associated with reduced responsiveness to BCR stimulation (Lee et al, 2022) and with anergy, whereby cells with a transgenic BCR specific for self-antigen expressed increased *Sdc1* transcripts (encoding CD138) (Sabouri et al, 2016). For PC, the expression of CD138 was found to play a role in their survival by binding factors including APRIL and IL-6 (Kimberley et al, 2009; McCarron et al, 2017). In keeping with this, mice deficient for *Sdc1* have fewer PC and reduced immunoglobulin (Ig) levels, particularly IgG (McCarron et al, 2017). CD138-linked HS constitutes approximately half of the HS on the surface of PC (McCarron et al, 2017) and the disruption of HS conformation also results in reduced PC survival (Reijmers et al, 2011), implying a causal relationship. However, *Sdc1*$^{-/-}$ mice were also found to have increased anti-glomerular basement membrane antibodies and changed IgG subclass distribution in a model of nephritis, the latter explained by changes in T-cell polarization (Rops et al, 2007).

The expression and function of SDC4 in B-lineage cells is comparatively poorly described. *Sdc4* transcripts are detectable in pre-B-cell-lines (Kim et al, 1994), and its expression in pro-B cells is regulated by the crucial B-cell transcription factor Pax5 (Medvedovic

---

[1]Department of Immunology, Monash University, Melbourne, VIC 3004, Australia. [2]Murdoch Children's Research Institute, Melbourne, VIC 3052, Australia. [3]National Heart and Lung Institute, Imperial College London, London, UK. [4]Australian Centre for Blood Diseases, Monash University, Melbourne, VIC 3004, Australia. [5]Blood Cells and Blood Cancer Division, The Walter and Eliza Hall Institute of Medical Research, Melbourne, VIC 3052, Australia. [6]Department of Medical Biology, The University of Melbourne, Melbourne, VIC 3052, Australia. [7]Olivia Newton-John Cancer Research Centre, Heidelberg, VIC 3084, Australia. [8]School of Cancer Medicine, La Trobe University, Heidelberg, VIC 3084, Australia. ✉E-mail: craig.mckenzie@monash.edu; isaak.quast@monash.edu

et al, 2011; Schebesta et al, 2007). SDC4 has been detected on the surface of mature B cells where it regulates the formation of dendritic processes (Yamashita et al, 1999) and *Sdc4*-deficient mice have fewer germinal center (GC) B cells in a model of collagen-induced arthritis (Endo et al, 2015). While these results suggest a role for SDC4 in B-cell activation, survival and/or differentiation, whether it influences B-cell development is unknown.

Here, we assess the expression and function of SDCs CD138 and SDC4 in mouse B cells to clarify their roles in development, activation, and differentiation. B-cell development was possible in the absence of SDCs, but we find tightly regulated expression patterns and an impact on specific developmental stages. In mature B cells, cytokines, BCR engagement and T-cell help dictate CD138 and SDC4 expression and loss. Moreover, soluble GAGs restrained sensitivity to BCR-mediated activation of naive B cells, while *Sdc4* deletion increased B-cell activation. These findings identify SDCs as stage-specific regulators of B-cell activation and survival with implications for the B-cell response to antigen.

## Results

### Expression of CD138 and SDC4 throughout B-cell development, maturation and differentiation

To understand the role of CD138 and SDC4 in B-cell development and function, we first characterized their expression during different B-cell development stages. BM and spleen from WT C57BL/6 mice were harvested and stained for CD138, SDC4, and cell surface molecules identifying B-cell subsets (Figs. EV1A,B and EV2) (Ng et al, 2020). The amount of both CD138 and SDC4, measured as the mean fluorescence intensity (MFI), increased on pro- and large pre-B cells compared to pre-pro-B-cell precursors, with CD138 expression maintained through the small pre-B stage but then lost on immature and recirculating B cells (Fig. 1A,B). After a strong induction in pro-B cells, SDC4 expression diminished during the progression to small pre-B cells, but was then restored on immature and recirculating B cells (Fig. 1A,B). CD138 and SDC4 are therefore differentially expressed during B-cell maturation with CD138 being lost and SDC4 gained during the final stages of BM B-cell development.

We next examined the expression of CD138 and SDC4 on mature splenic B-cell subsets, resolving populations of immature/transitional (IgD⁻ IgM⁺), naive (IgD⁺ IgM⁺), GC (IgD⁻ FAS⁺), class-switched B cells (IgM⁻ IgD⁻) and PC (FSC-A^hi CD19^int IgD⁻ CD98⁺) (Figs. 1C and EV2A,B). Consistent with prior reports (Tung et al, 2006), little to no expression of CD138 was detected on naive B cells (Fig. 1D). Around one-sixth (16%) of isotype class-switched B cells showed some expression of CD138, albeit at much lower levels than PC (Fig. 1C,D). SDC4 was expressed at high amounts on immature and naive B cells and ~30–40% of class-switched B cells, with GC B cells and PC showing very low or no expression (Fig. 1D). Thus, in mature B-lineage cells, CD138 is primarily expressed on PC while SDC4 is high on naive B cells, lost upon activation and shows a bimodal expression pattern on class-switched memory B cells.

The bimodal expression on class-switched B cells prompted the question of whether SDC4 was associated with memory B-cell subsets. To study SDC expression on MBC, we immunized C57Bl/6 mice with alum-adjuvanted (4-hydroxy-3-nitrophenyl)acetyl-conjugated keyhole limpet hemocyanin (NP-KLH) and analyzed

NP-specific cells 3 weeks thereafter. MBC were identified as FAS^int CD38⁺IgM⁻IgD⁻ NP-binding cells (Quast, 2024; Ridderstad and Tarlinton, 1998) (Fig. EV2C) and MBC subsets defined by the expression of CD44, CD62L, CD80 and PD-L2. CD44 is gradually re-expressed following GC exit (Hanson et al, 2023) and CD44⁺CD62L⁺ MBC transcriptionally resemble CD80⁻PD-L2⁻ MBC (Laidlaw et al, 2020), a subset that has been shown to re-seed GC upon rechallenge, while CD80⁺PD-L2⁺ MBC preferentially differentiate into PC (Zuccarino-Catania et al, 2014). NP-binding MBC uniformly expressed CD44 (Fig. EV2D,E), and ~25% of total MBC expressed SDC4, irrespective of CD62L expression (Fig. 1E,F). In contrast, twice as many CD80⁻PD-L2⁻ MBC expressed SDC4 compared to CD80⁺PD-L2⁺ MBC (Fig. 1F). These data show that while SDC4 remains absent in most B cells following MBC differentiation, subsets that respond distinctly following rechallenge differ in SDC4 expression.

### CD138 and SDC4 regulate specific stages of B-cell development

Having established the expression patterns of CD138 and SDC4, we next aimed to gain insights into their functional roles during B-cell development. To this end, we used CRISPR–Cas9 genome editing to abrogate *Sdc1* and *Sdc4* in the haemopoietic compartment. BM from mice with widespread, transgene-encoded, CAG promotor-driven expression of Cas9 and GFP (Chu et al, 2016) was transduced with a lentiviral vector (Koike-Yusa et al, 2014) expressing a blue fluorescent protein (BFP) reporter and single guide RNAs (sgRNAs) to either *Sdc1* or *Sdc4* (Fig. 2A). Lentiviral transduced BM was then used to reconstitute lethally irradiated WT C57Bl/6 recipient mice. This produced chimeric mice in which lentiviral transduced, sgRNA-expressing cells were identified as being GFP⁺BFP⁺ and non-gene-targeted cells as GFP⁺BFP⁻ (Fig. EV3A). At 8 weeks post reconstitution, mice were immunized with NP-KLH to assess B-cell activation and differentiation. To allow the identification of PC in CD138 targeted and non-targeted mice, we used high CD98 expression (Robinson et al, 2023; Tellier et al, 2016) combined with high forward scatter and intermediated CD19 expression (Fig. EV3B). When applied to non-CD138 targeted mice, this resulted in a PC population of which >85% expressed CD138 (Fig. EV3B). To further validate this approach, we applied the gating strategy to mice expressing red fluorescent protein (tdTomato) under the control of the *Prdm1* locus (Robinson et al, 2023; Robinson et al, 2022), encoding the PC transcription factor BLIMP1 (Kallies et al, 2004; Shapiro-Shelef et al, 2003). Over 90% of cells in the final gate expressed tdTomato, confirming PC identity (Fig. EV3C). Using this strategy to assess CD138 on splenic PC and separately SDC4 on naive B cells 9 weeks post BM reconstitution confirmed successful gene deletion in a sizeable fraction of GFP⁺BFP⁺ cells (Fig. 2B,C). Beyond validating our experimental system, this analysis also revealed immediately that although CD138 was found to have crucial roles in PC survival (McCarron et al, 2017), ~half of splenic PC in *Sdc1*-targeted cells lacked CD138 expression, and CD138 was thus not absolutely required for PC generation and survival in the short-term (Fig. 2B).

We next investigated the impact of CD138 deletion in B-cell development and activation. We identified B-cell maturation subsets in the BM by flow cytometry (Ding et al, 2024) (Fig. EV4A) and assessed changes in the ratio of BFP⁺ to BFP⁻ cells as an

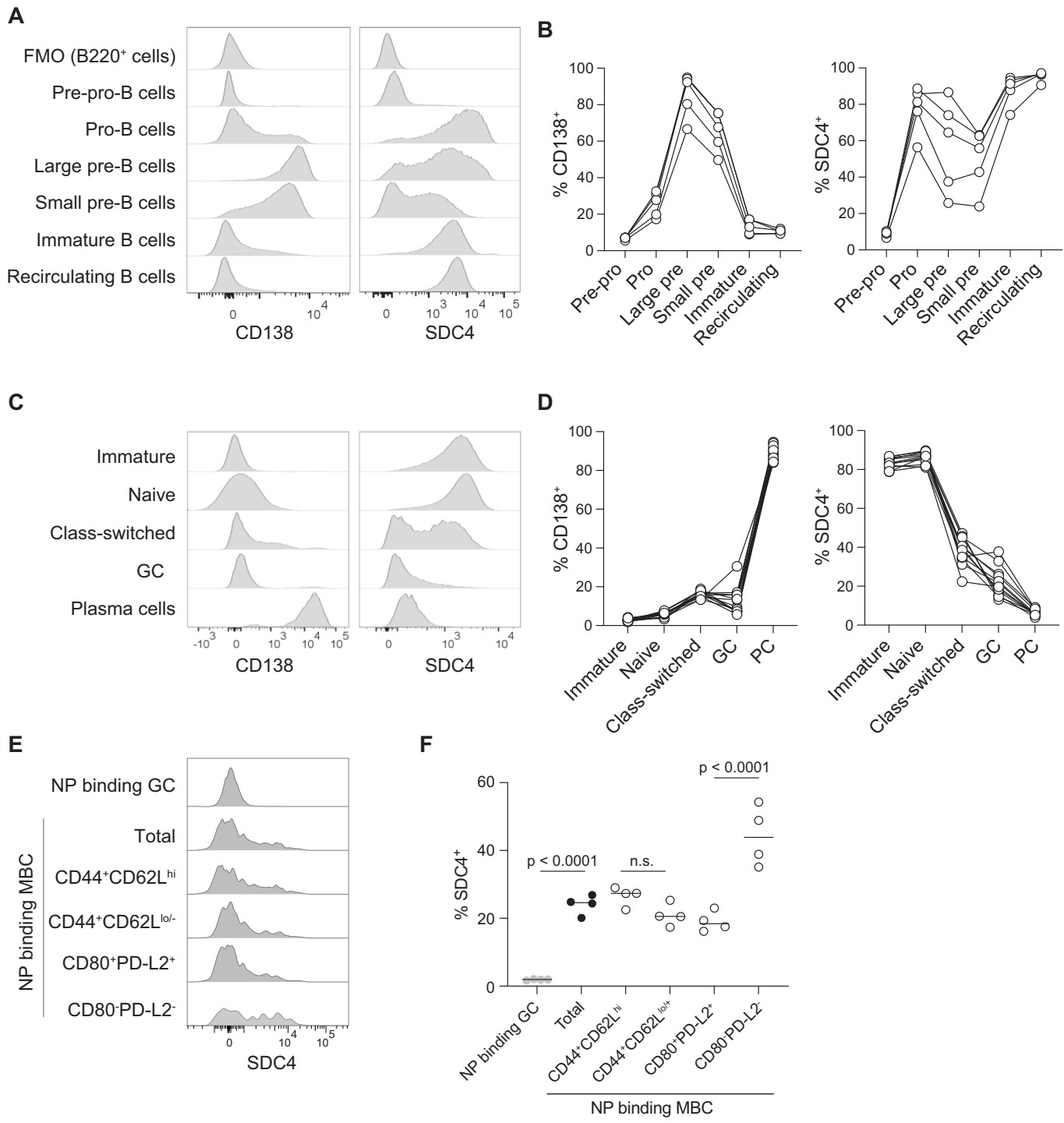

**Figure 1. CD138 and SDC4 are differentially expressed during B cell development and maturation.**

(A) Representative histograms of CD138 and SDC4 expression on developing B-cell subsets in murine BM analyzed by flow cytometry. FMO, fluorescence minus one control (not stained for either CD138 or SDC4). (B) Proportion of CD138$^+$ or SDC4$^+$ BM B-cell subsets. (C) Representative histograms of CD138 and SDC4 expression on splenic B cells and PC. (D) The proportion of B cells expressing CD138 and SDC4 among mature B-cell subsets. (E) SDC4 expression on splenic NP-specific GC, MBC, and MBC subsets 3 weeks post NP-KLH immunization and (F) statistical analysis. Data show one experiment with 4–5 biological replicates with lines in (B, D) connecting values of individual mice. Statistical analysis in (F) by one-way ANOVA with Šídác multiple comparisons test. Source data are available online for this figure.

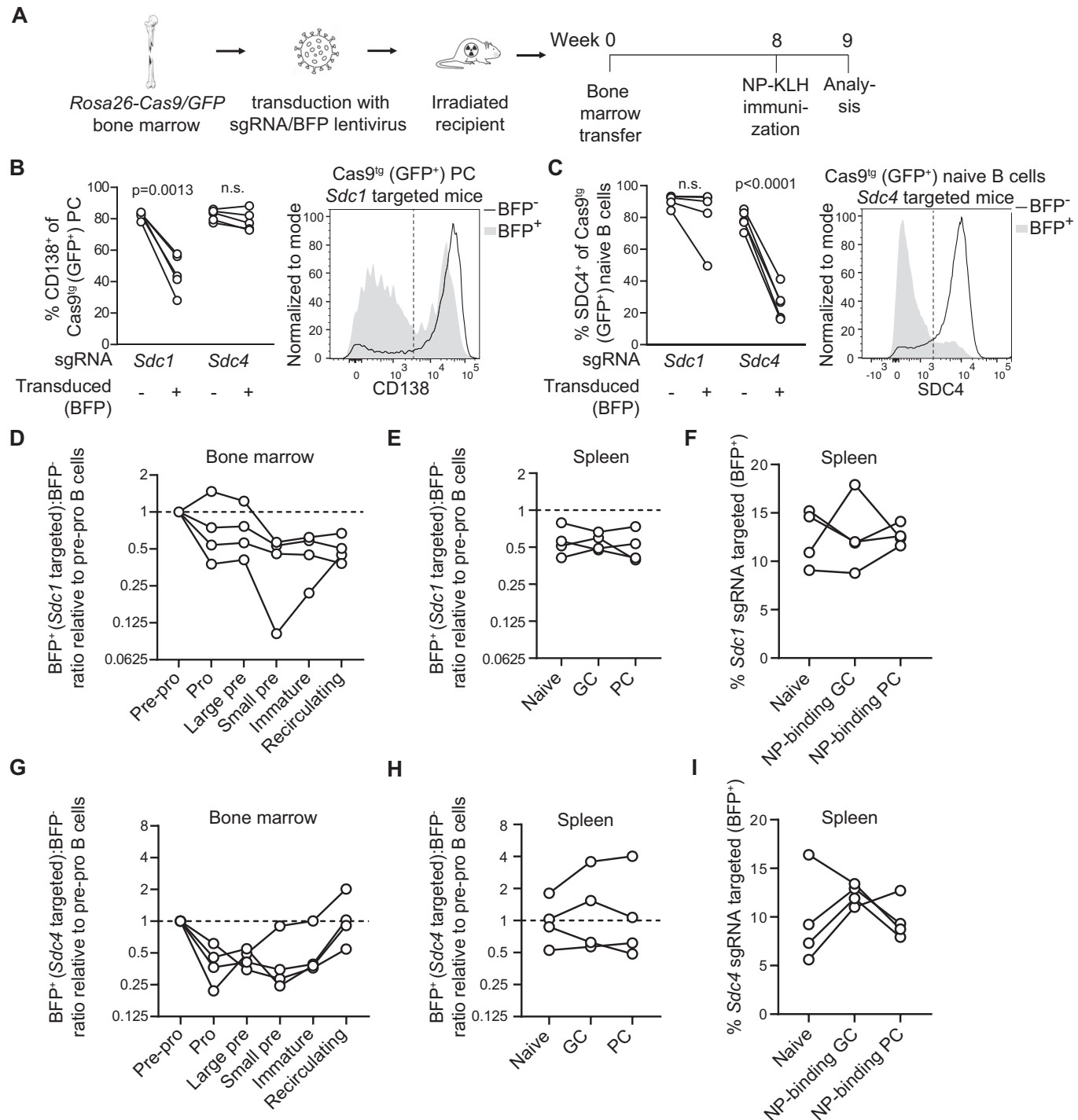

**Figure 2. Ablation of CD138 and SDC4 shows a non-essential role in B-cell development.**

(A) Schematic of CRISPR–Cas9 genome editing strategy. Lentiviral vectors encode BFP and sgRNAs targeting either *Sdc1* or *Sdc4*. Transduced BM was transferred into lethally irradiated wild-type recipients, and 8 weeks after BM transfer, recipients were immunized i.p. with alum-adjuvanted NP-KLH. One-week post-immunization, spleen and BM were harvested for analysis by flow cytometry with cells reconstituted from Cas9 BM identified by GFP expression. (B) The proportion of GFP$^+$ splenic PC expressing CD138 in mice reconstituted with lentiviral transduced BM (encoding sgRNA as indicated) and exemplary CD138 expression. (C) The proportion of GFP$^+$ splenic naive B cells expressing SDC4 in mice reconstituted with lentiviral transduced BM (encoding sgRNA as indicated) and exemplary SDC4 expression. (D–F) Representation of lentiviral transduced (BFP$^+$, expressing sgRNA targeting *Sdc1*) cells among total GFP$^+$ developing (D) and mature (E) B-cell subsets from the BM and spleen, respectively, and (F) the proportion of BFP$^+$ cells among splenic naive B cells as well as NP-binding GC B cells and PC. (G–I) Are as (D–F), respectively, but for mice reconstituted with BM targeted for *Sdc4* deletion. Data show biological replicates ($n = 4$ BM-reconstituted mice) with lines connecting individual mice. Statistical analysis by paired $t$ test. Source data are available online for this figure.

indicator of successful progression, with the ratio of the earliest committed B-cell precursor, pre-pro-B cells, set to 1 for each mouse. The proportion of small pre-B cells targeted for *Sdc1* was reduced by approximately 50%, which remained stable throughout subsequent B-cell maturation stages in BM, including recirculating B cells (Fig. 2D). Similarly, *Sdc1* targeting did not influence the representation of these cells among splenic GC and PC compared to naive B cells when total (Fig. 2E) or NP-binding GC B cells and PC (surface immunoglobulin expression by early PC allows detection of antigen-specific cells (Blanc et al, 2016; Bortnick et al, 2012)) were compared (Fig. 2F), arguing against a role of CD138 for B-cell activation or PC differentiation. It is important to note that our data do not allow conclusions about the potential role of CD138 in regulating PC lifespan as the total PC pool is constantly replenished and long-lived PC make up only a small fraction of all PC (Robinson et al, 2023; Robinson et al, 2022). However, at the level of analysis conducted here, and aside from a possible drop in small pre-B cells, CD138 had no or only a minor role in B-cell development and was dispensable for antigen-induced activation and PC differentiation.

*Sdc4*-targeted B cells were reduced during the pre-pro to pro-B-cell transition (Fig. 2G), coinciding with the upregulation of SDC4 at this developmental stage (Fig. 1A,B). This loss was counteracted by the gradually increased representation of *Sdc4*-targeted B cells during the final stages of development (Fig. 2H). Last, total GC B cells and PC were largely unaffected in *Sdc4*-targeted mice while NP-binding GC B cells showed (except for one mouse) a trend towards increased representation compared to naive B-cell precursors (Fig. 2I). These data indicated that SDC4 promoted B-cell development at the pro-B stage but subsequently limited progression to mature circulating B cells.

A key signal required for B-cell lymphopoiesis is BM stromal-cell-derived IL-7 (Cordeiro Gomes et al, 2016; Miller et al, 2002) and enzymatic removal of HS from B-cell precursors was shown to reduce IL-7 binding and IL-7-induced proliferation (Borghesi et al, 1999). We therefore investigated if SDC deletion affects IL-7-driven B-cell development in vitro (Holl et al, 2010). BM from *Sdc1* or *Sdc4*-targeted mice was cultured in the presence of IL-7, and the composition of B-cell precursors was analyzed on days 0, 4, and 8 (Fig. EV4B). The total number of GFP$^+$ B-cell precursors (pre-pro to immature B cells) increased more than tenfold, dominated by large and small pre-B cells (Fig. EV4C,D). To compare the effect of SDC deletion on B-cell differentiation in vivo and in vitro, we analyzed the representation of gene-targeted (GFP$^+$BFP$^+$) cells in each precursor population relative to pre-pro-B cells at the start of the culture. Similar to in vivo results, *Sdc1*-targeted cells gradually decreased with each consecutive differentiation step and over time (Fig. EV4E), while *Sdc4*-targeted cells showed a drop from pre-pro- to pro-B cells followed by a low but mostly steady representation (Fig. EV4F).

Collectively, our results show that B-cell development can occur in vivo in the absence of CD138 or SDC4, but both SDCs affect specific stages of B-cell development, and in vitro data suggest a role in supporting IL-7-mediated B-cell differentiation.

## BCR, cytokine, and CD40 signaling impact SDC expression by B cells

Based on the above results and combined with previous reports showing expression of CD138 to be associated with antigen-induced

B-cell anergy (Lee et al, 2022), we speculated whether SDC expression was responsive to activation stimuli such as BCR stimulation, the receipt of T-cell help or T-cell-derived cytokines involved in B-cell activation, proliferation and survival. To study these stimuli, we used B cells from mice transgenic for a BCR specific for hen egg lysozyme (HEL, mice known as SW$_{HEL}$ mice) (Phan et al, 2003). These mice were additionally *Rag1* deficient, resulting in the absence of T cells and the prevention of BCR recombination during B-cell development, thus generating a source of uniformly naive B cells with defined specificity (Quast et al, 2022). To reveal any relationship between cell division and changes in SDC expression, SW$_{HEL}$ splenocytes were stained with the division tracking dye cell trace violet (CTV) before in vitro cell culture in the presence or absence of HEL antigen and IL-21, IL-4 or anti-CD40 for 3 days. As expected, naive SW$_{HEL}$ B cells lacked CD138 but expressed high levels of SDC4. BCR stimulation with HEL caused CD138 upregulation, particularly when combined with IL-21, while IL-4 and anti-CD40 prevented such induction of CD138 expression (Fig. 3A). Strikingly, in the absence of BCR stimulation, neither IL-4 and IL-21 nor anti-CD40 influenced CD138 or SDC4 expression (Fig. 3A,B). Moreover, the antigen-induced reduction in SDC4 expression was counteracted by IL-4, anti-CD40 and, to a lesser extent, IL-21 (Fig. 3B). This BCR signaling-induced regulation was replicated in polyclonal B cells from C57Bl/6 mice stimulated with anti-Igκ/λ (Fig. EV5A). Analysis of SDC expression across CTV division peaks showed that antigen-induced CD138 upregulation occurred independently of cell division and was followed by a gradual loss with consecutive divisions (Fig. EV5B). PC express high levels of CD138 but the transient CD138 expression induced by antigen stimulation is not associated with the initiation of the PC program, which only starts after >5 cell divisions (Dvorscek et al, 2022; Hasbold et al, 2004; Scharer et al, 2018). For SDC4, antigen stimulation initiated a gradual loss to an extent dependent on cytokines and T-cell help (Fig. EV5B), while in the absence of antigen, these stimuli had again no or very little effect on CD138 and SDC4 expression even in cells that had undergone multiple rounds of division (Fig. EV5C). To determine whether activation-induced changes in SDC expression on B cells were observed in vivo, we transferred GFP$^+$ SW$_{HEL}$ B cells alongside ovalbumin (OVA)-specific OT-II CD4 T cells into WT recipients and immunized intraperitoneally (i.p.) with HEL fused to the peptide of OVA (OVA$_{pep}$) recognized by OT-II T cells (Barnden et al, 1998) adsorbed on alum adjuvant (as previously described (Quast et al, 2022)). Three days post-transfer and immunization, CD138 expression on SW$_{HEL}$ B cells (identified as CD19$^+$GFP$^+$ cells, Fig. EV5D) were increased relative to endogenous naive and GC B cells (Fig. 3C). This was consistent with our observation of antigen-induced CD138 expression on SW$_{HEL}$ B cells in vitro (Fig. 3A). SDC4 expression on the now activated SW$_{HEL}$ B cells was reduced (Fig. 3D), again consistent with the observed reduction of SDC4 on B cells activated in vitro (Fig. 3A). Thus, CD138 and SDC4 are regulated by cognate antigen and further modulated by the T-cell-derived signals IL-4, IL-21, and CD40 ligation, suggesting a role for SDCs in regulating B-cell activation.

## Soluble heparin, HS, and CS promote B-cell survival in response to cognate antigen

HS on SDCs can be cleaved by the enzyme heparinase to form soluble HS, which can mediate diverse functions independently of the protein to which it was attached (Simon Davis and Parish,

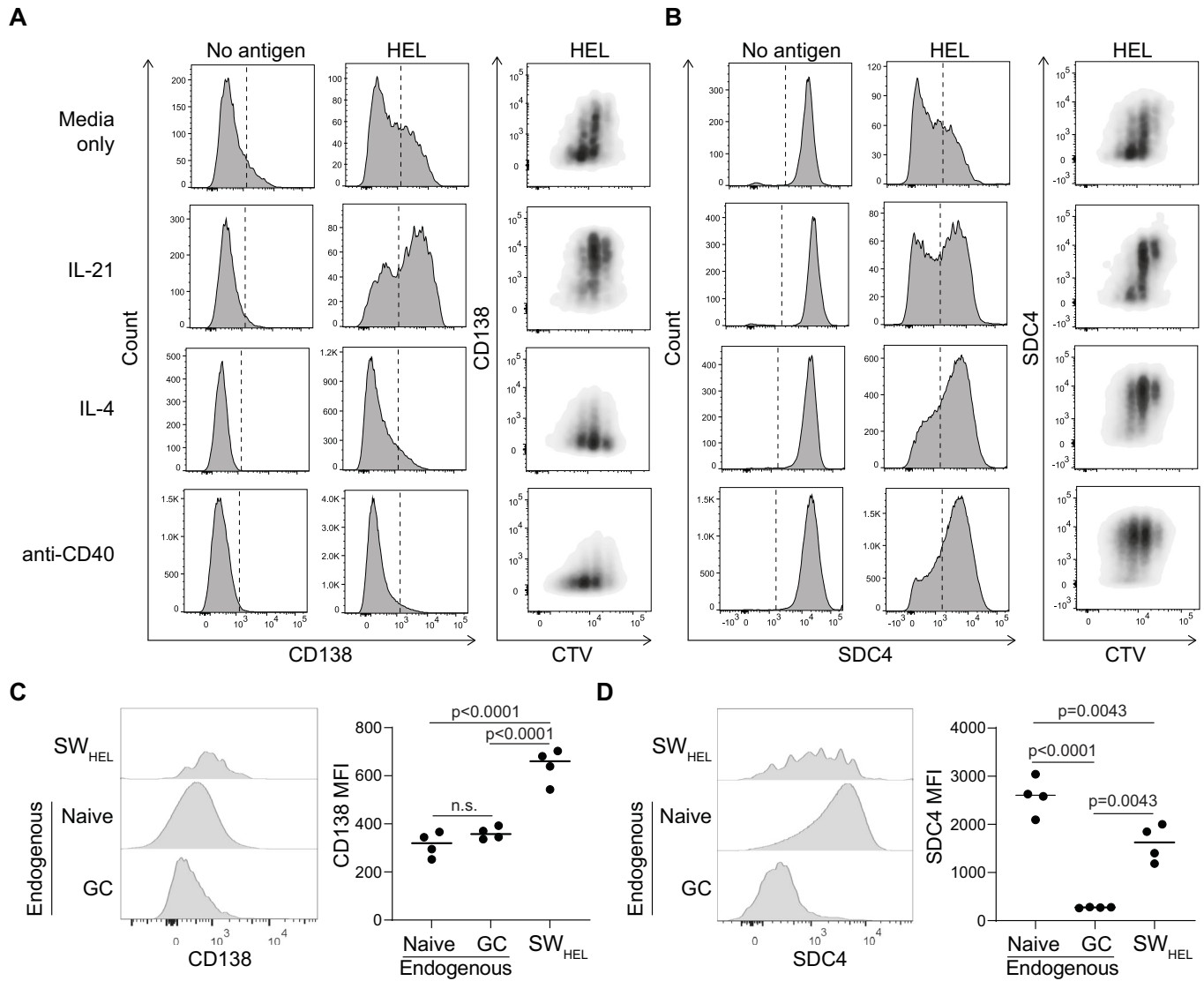

**Figure 3. Regulation of SDC expression by cognate antigen, cytokines, and T-cell help.**

Representative histograms and density plots of (**A**), CD138 and (**B**), SDC4 amounts on CTV-labeled RAG-1$^{-/-}$ SW$_{HEL}$ B cells stimulated with or without HEL in the presence of recombinant IL-21, IL-4 or anti-CD40 antibody for 3 days followed by analysis by flow cytometry. (**C, D**) Recipients of SW$_{HEL}$ B cells and OT-II T cells were immunized i.p. with alum-adjuvanted HEL-OVA$_{pep}$ and spleens analyzed 3 days thereafter. Representative histogram and MFI of (**C**) CD138 and (**D**) SDC4 expression on splenic RAG-1$^{-/-}$ SW$_{HEL}$ B cells and endogenous naive and GC B cells. Representative data from two experiments. Data in (**C, D**) show individual mice ($n = 4$) with statistical analysis by one-way ANOVA with Tukey's multiple comparisons test. Source data are available online for this figure.

2013). For example, soluble HS can act as a ligand of TLR4 (Johnson et al, 2002) and cleavage of HS from the surface of endothelial cells promotes leukocyte migration through the extracellular matrix (Mayfosh et al, 2019). Heparinase expression has been detected in mature B cells (Laskov et al, 1991), but the potential role of free HS in altering B-cell activation or survival is not known. Given the dynamic regulation of GAG-containing SDCs, we sought to determine the impact of HS, CS and heparin, a naturally occurring GAG and ubiquitous clinical anticoagulant with a similar structure to HS (Qiu et al, 2021), on B cells. We first tested the effect of high-dose (250 IU/mL) heparin on the survival of B cells in vitro. Unexpectedly, the addition of heparin potently induced B-cell proliferation, more so than the combination of anti-

CD40 and IL-4 (Fig. EV6A,B), and did so without the activation-induced upregulation of FAS (Fig. EV6C). Thus, at high concentrations, heparin acted as a B-cell mitogen. Next, we tested the effect of soluble heparin on naive and antigen-stimulated B cells at a concentration equal to that found in the blood of heparin-treated hemodialysis patients (2.5 IU/mL) (Tovar et al, 2013) and compared this to HS and CS. At this concentration, heparin, HS, and CS had no impact on the number of B cells surviving in culture in the absence of antigen (Figs. 4A and EV6D), but heparin, and to a lesser extent HS and CS, significantly increased the number of SW$_{HEL}$ B cells by day 3 of culture in the presence of HEL antigen (Fig. 4A). As such, heparin, HS and CS resulted in increased B-cell numbers within each cell division, with heparin also slightly

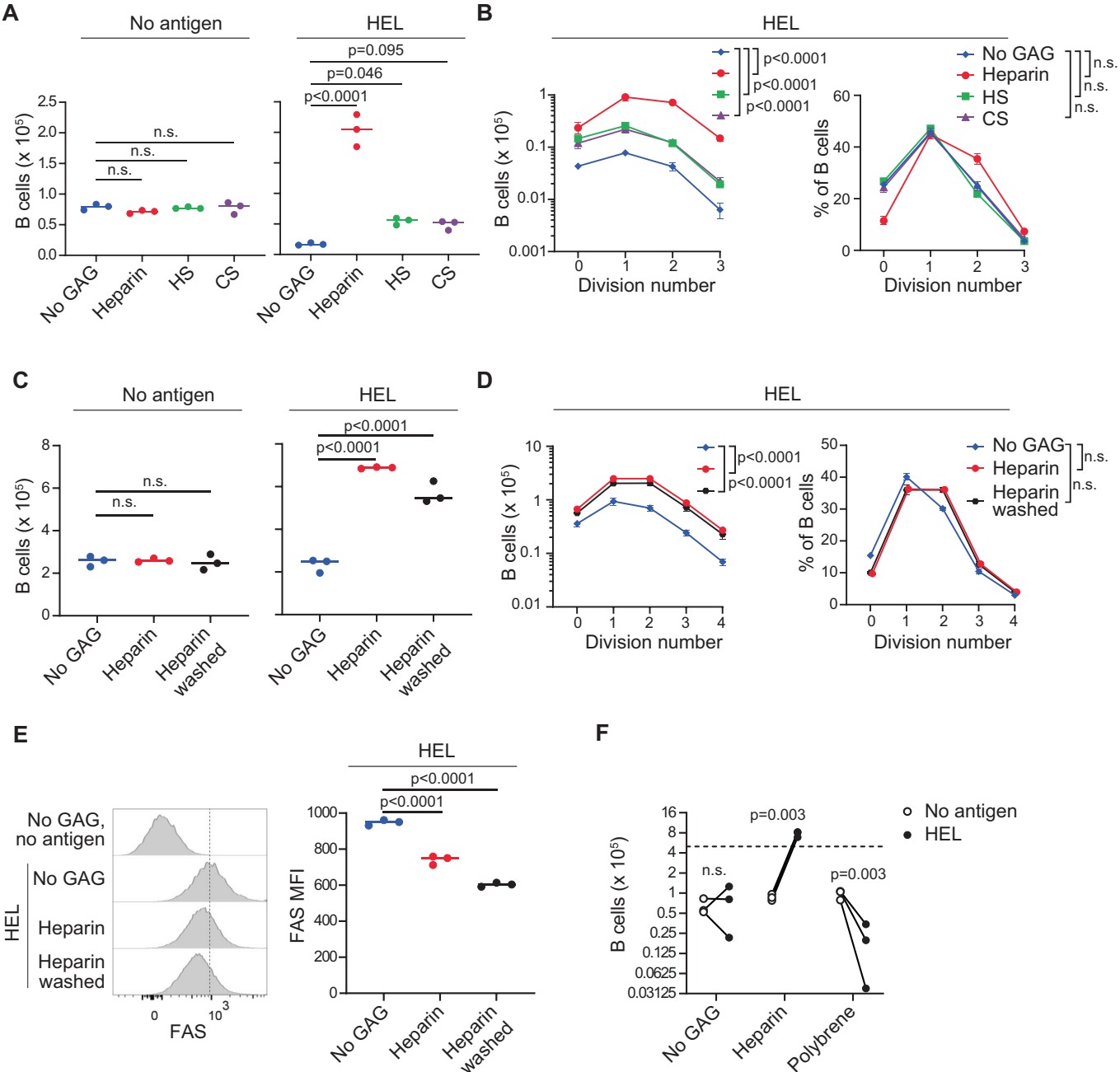

**Figure 4. Soluble GAGs increase survival of B cells activated by cognate antigen.**

(A) Number of RAG-1$^{-/-}$ SW$_{HEL}$ B cells after 3 days of culture with or without HEL in the presence of heparin, HS, or CS and (B) further analyzed per cell division. (C) Number of RAG-1$^{-/-}$ SW$_{HEL}$ B cells stimulated with or without HEL and cultured with heparin-supplemented media compared to B cells that had been briefly incubated with heparin on ice and then washed before 3 days of culture. (D) Number and proportion of RAG-1$^{-/-}$ SW$_{HEL}$ B cells per division from data shown in (C). (E) Representative histogram and MFI of FAS expression on cells from (C). (F) Number of RAG-1$^{-/-}$ SW$_{HEL}$ B cells after 3 days of culture with or without HEL in the presence of heparin or polybrene (lines connect data from individual mice ($n = 3$), dotted line indicates B-cell number at the start of culture). Data in (A–E) show technical triplicates, representative of two experiments with error bars indicating standard deviation. Statistical analysis in (A, C, E) by one-way ANOVA with Tukey's post test and in (B, D) by two-way ANOVA with Tukey's multiple comparisons test. Statistical analysis in (F) by paired $t$ test. Source data are available online for this figure.

increasing the rate of cell division (Fig. 4B). As heparin has been demonstrated to bind a variety of B-cell surface molecules, including Ig (Aricescu et al, 2002; Levy et al, 1981), we tested if short-term heparin exposure was sufficient to promote B-cell proliferation and survival. B cells cultured with heparin for 30 min followed by washing and re-seeding in new culture medium without heparin for 3 days survived and proliferated similarly to B cells provided with continuous soluble heparin in culture media (Figs. 4C,D and EV6E). Of note, the survival of SW$_{HEL}$ B cells in the absence of antigen varied between experiments (Fig. 4A,C) but

the effect of heparin was highly consistent across experiments. Heparin exposure again resulted in lower expression of FAS, indicating reduced activation in response to antigen (Hahne et al, 1996) (Fig. 4E). Lastly, we assessed if polybrene, a long, positively charged molecule capable of neutralizing the anticoagulant properties of heparin (Montalescot et al, 1990), would have the opposite effect. Indeed, polybrene reduced B-cell numbers in response to HEL, had no impact in the absence of antigen (Fig. 4F), and counteracted the effect of heparin in a dose-dependent manner (Fig. EV6F). Collectively, these data show that soluble GAGs can modulate B-cell activation, increase B-cell survival following antigen-mediated activation, and at higher concentrations, act as B-cell mitogen.

### BCR-mediated B-cell activation is inhibited by heparin and promoted by heparinase

As B cells stimulated in vitro with high-affinity antigen can undergo rapid activation-induced cell death (AICD) (Wensveen et al, 2016), we examined whether this was influenced by heparin. The number of viable $SW_{HEL}$ B cells was markedly reduced after 30 min of high-affinity antigen (HEL) stimulus, and this was partially rescued by heparin (Fig. 5A). Those B cells surviving antigen stimulation expressed the activation marker CD69 and exhibited a striking reduction in SDC4 expression, with both effects counteracted by heparin (Fig. 5A). Based on this reduced B-cell activation in the presence of heparin, we hypothesized that removing HS from the surface of B cells using heparinase could have the opposite effect. To investigate this, we used HEL containing three mutations (known as $HEL^{3x}$) that lower its affinity for $SW_{HEL}$ BCR from $K_a = 2 \times 10^{10}$ to $\sim 1 \times 10^7 \, M^{-1}$, resulting in sub-maximal BCR activation (Chan et al, 2012; Dvorscek et al, 2024). As expected, stimulation of $SW_{HEL}$ B cells with $HEL^{3x}$ for 30 min did not induce AICD but resulted in B-cell activation as indicated by upregulation of CD69 and downmodulation of SDC4 (Fig. 5B). Heparinase treatment of $SW_{HEL}$ B cells before exposure to $HEL^{3x}$ did indeed augment expression of CD69, but SDC4 and AICD remained unchanged (Fig. 5B). We concluded from these data that HS, including those attached to B-cell surface molecules, could limit B-cell activation. With naive B cells expressing high levels of SDC4, this suggested that *Sdc4* deletion would promote B-cell activation. To test this proposal, we assessed if SDC4 influenced B-cell activation by quantifying the phosphorylation of S6, a central regulator of proliferation and metabolism activated in response to cytokines, T-cell help and BCR signaling (Dvorscek et al, 2022; Luo et al, 2023; Zotos et al, 2021). In line with S6 phosphorylation depending on mTOR signaling (Battaglioni et al, 2022), the presence of the mTOR inhibitor rapamycin prevented S6 phosphorylation in response to BCR stimulation (Fig. EV6G). B cells from mice targeted for *Sdc4* deletion by CRISPR–Cas9 (as described in Fig. 3) had increased baseline amounts of phosphorylated S6 in naive B cells, which was further increased following in vitro activation through BCR stimulation for 30 min using anti-Igκ/λ (Fig. 5C). To analyze BCR proximal events, we FACS sorted splenic GFP+BFP- (non-targeted) and GFP+BFP+ (lentivirally targeted) cells from BM-reconstituted mice and stimulated them for 5 or 30 min with anti-Igκ/λ, followed by analysis of Syk phosphorylation by flow cytometry. B cells from *Sdc4* but not *Sdc1* sgRNA-targeted B cells showed increased baseline Syk

phosphorylation compared to non-targeted B cells from the same mouse, while pSyk in response to anti-Igκ/λ-mediated BCR stimulation was comparable for all conditions (Fig. EV6H,I). Combining these results with our earlier observations on the diametric effects of heparin and heparinase on B-cell activation suggests SDC4 expressed on mature naive B cells restricts B-cell activation and does so at least in part through attached HS.

## Discussion

This study provides a detailed assessment of CD138 and SDC4 expression and function throughout B-cell development, activation, and differentiation. We show that PC can be readily generated in the absence of CD138 in vivo and provide novel insights into the role of SDC4 in B-cell activation. SDCs are known for their role in adhesion as well as growth factor and cytokine binding, functions that imply a regulatory rather than essential role for lymphocyte function. In agreement, neither molecule was required for B-cell development in vivo with only minor changes on the progression from stem cells to naive, mature B cells. The most prominent effect was a reduction in *Sdc4*-targeted B cells during the early stages of B-cell development. This is in line with HS as a ligand for pre-BCR (Bradl et al, 2003) where it promotes ERK1/2 phosphorylation following in vitro pre-BCR ligation (Milne et al, 2008) and with a role of HSPG in IL-7-mediated B-cell development (Borghesi et al, 1999).

CD138 is expressed on anergic B cells (Sabouri et al, 2016), and the loss of SDC4 following GC B-cell formation reported here suggests a role for proteoglycans in B-cell activation. Investigating this possibility, we identify CD138 as a molecule induced by antigen stimulation and that this effect is prevented by IL-4 and anti-CD40. This in turn may explain CD138 upregulation on B cells exposed to abundant self-antigen (Sabouri et al, 2016) and highlights that CD138 expression does not necessarily identify functionally anergic cells but rather those cells that recently encountered antigen but have not, or not yet, received cognate T-cell help. The expression pattern of SDC4 was reciprocal to that of CD138 in that its expression was downregulated following antigen encounter and sustained by IL-4 and in particular by anti-CD40. These signals did not completely block SDC4 down-regulation but rather delayed the extent of loss during consecutive cell divisions, explaining why GC B cells that undergo extensive T-cell-dependent proliferation express no or very low levels of SDC4. In addition to being regulated by cytokines and T-cell help, SDC4 may itself regulate the sensitivity of B cells to cytokine signaling. As such, IL-21, a key driver of B-cell proliferation before (Dvorscek et al, 2022) and within (Linterman et al, 2010; Luo et al, 2023; Petersone et al, 2023; Quast et al, 2022; Zotos et al, 2021) GC has been demonstrated to bind HS on the surface of B cells, and that the loss of HS on GC B cells reduces the capacity to bind IL-21 (Chen et al, 2023). Our data suggest that perhaps SDC4 mediates binding of IL-21 through HS/CS and the reduction of SDC4 on GC B cells reflects the reduction of HS observed on GC B cells (Chen et al, 2023).

A key result of our study is the independence of homeostatic and antigen-induced PC from CD138. This was surprising given a previous study reported that while PC generation was not affected by CD138, PC lacking CD138 due to genetic deletion or to their

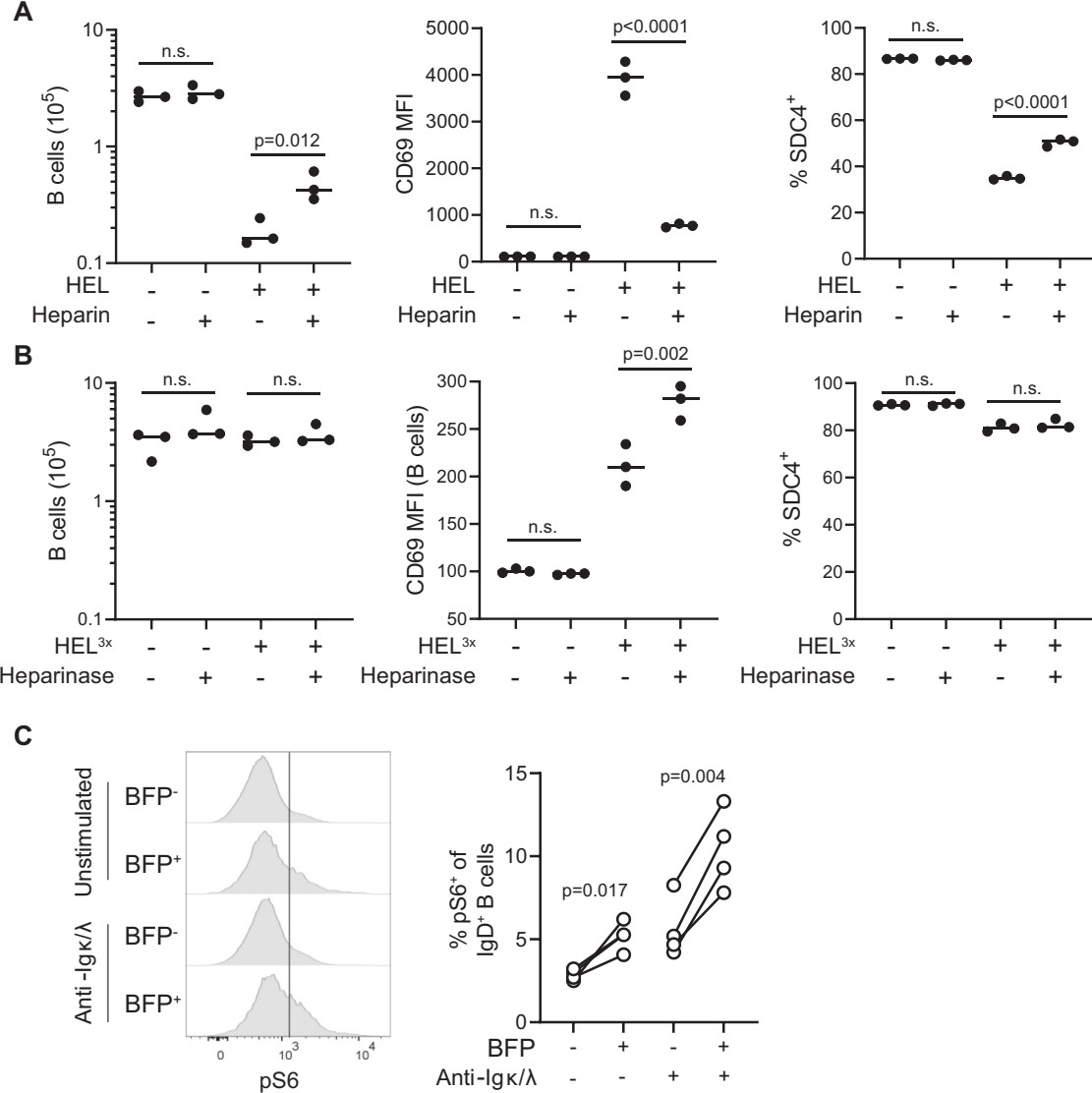

**Figure 5. GAG and SDC4 reduce activation of B cells stimulated with cognate antigen.**

(A) Number of RAG-1$^{-/-}$ SW$_{HEL}$ B cells and their expression of CD69 and SDC4 after 30 min incubation with or without HEL and in the presence or absence of heparin or (B), with or without heparinase treatment. (C) Representative histograms and proportion of naive B cells with phosphorylated S6 (at serine$_{235/236}$) among B cells derived from mice reconstituted with BM targeted for *Sdc4* deletion as shown in Fig. 3. Cells were either left untreated or stimulated for 30 min with anti-Igκ and anti-Igλ and S6 phosphorylation analyzed by flow cytometry, distinguishing Cas9$^{tg}$ (GFP$^+$) lentiviral transduced (*Sdc4* targeted, BFP$^+$) and non-transduced (BFP$^-$) B cells. Data in (A, B) show technical replicates and (C) biological replicates ($n = 4$ mice) representative of two experiments. Statistical analysis in (A, B) by *t* test and in (C) by paired *t* test. Source data are available online for this figure.

immature state were prone to cell death as a consequence of reduced responsiveness to APRIL and IL-6 (McCarron et al, 2017). However, the same study found total IgG but not IgM titers to be reduced in *Sdc1*$^{-/-}$ mice and other groups reported altered isotype distribution, rather than reduced antibody titers (Rops et al, 2007). So while there is an agreement that CD138 is not involved in PC generation, its contribution to short-term survival appears to vary dependent on the specific circumstances investigated. Nevertheless, CD138's role in cell adhesion (Beauvais et al, 2009; Koda et al, 1985) and as a co-receptor for APRIL (Ingold et al, 2005; Kimberley et al, 2009), both crucial aspects for PC survival (Belnoue et al, 2008; Tarlinton et al, 2023), combined with the high CD138

expression found on long-lived PC (Robinson et al, 2023), makes involvement in regulating PC longevity likely. But rather than being necessary for PC survival, CD138 may influence the average lifespan of PC by modulating the bioavailability of survival factors (Simons and Karin, 2024).

Our study also provides novel insights into the regulation of B-cell activation by revealing SDC4 and GAGs as negative regulators. As such, exposure of B cells to GAGs reduced antigen-induced activation and increased survival following antigen encounter. For heparin, this occurred at concentrations used for treating patients, raising possible implications for immune cell activation, in particular because the effect was equally potent upon

short-term heparin exposure as expected to occur in vivo due to its short biological half-life (Bara et al, 1985). While our results do not directly show that SDC4-linked heparins can also mediate this effect, abrogation of SDC4 expression had the opposite effect to GAG supplementation, with B cells showing increased activation as measured by intracellular p-S6. This is also in line with a direct role for SDC4 in BCR-mediated activation, as suggested by the proposed interaction of HS with BCR (Gomez Toledo et al, 2021; Levy et al, 1981). Moreover, reduced S6 phosphorylation is consistent with a prior study that reported muscles isolated from sedentary *Sdc4*-deficient mice have increased Akt, mTOR and S6 phosphorylation compared to controls (Ronning et al, 2020). Phosphorylation of these PI3K-Akt signaling molecules is critical for B-cell activation and survival (Del Pino-Molina et al, 2021; Limon and Fruman, 2012) and SDC4 may therefore regulate signaling downstream from the BCR via GAGs and SDC4 cytoplasmic domains acting directly on intracellular signaling molecules. Indeed, in epithelial cells, cytoplasmic domains of SDC4 have been proposed to cluster into lipid rafts where they can act directly to restrict mTOR1 activity, thereby reducing S6 phosphorylation (Partovian et al, 2008). T-cell help via cytokines or CD40 signaling stimulated or retained SDC4 expression over consecutive cell divisions following naive B-cell activation, and it is conceivable that this mechanism facilitates pre-GC B-cell expansion. With naive B cells expressing high levels of SDC4 and MBC relatively fewer or none, SDC4 could also contribute to the intrinsic difference of naive and MBC in response to BCR stimulation (Akkaya et al, 2020; Tangye et al, 2003).

In conclusion, our data show that SDCs are dynamically regulated throughout B-cell development and in response to cognate antigen, but are dispensable for PC formation and at least short-term survival. Furthermore, we identify GAGs as modulators of B-cell activation and add SDC4 as a novel negative regulator of BCR signaling.

# Methods

### Reagents and tools table

| Antibody | Source | Identifier |
|---|---|---|
| B220-A647 (clone RA3-6B2) | WEHI Antibody Facility | N/A |
| B220-PE-Cy7 (clone RA3-6B2) | BD Biosciences | Cat# 552772 |
| B220-BV711 (clone RA3-6B2) | BioLegend | Cat#103255 |
| CD138-APC (clone 281-2) | BD Biosciences | Cat# 558626 |
| CD138-BV650 (clone 281-2) | BD Biosciences | Cat# 564068 |
| CD138-PE-Cy7 (clone 281-2) | Biolegend | Cat# 142514 |
| CD19-BUV737 (clone 1D3) | BD Biosciences | Cat# 612781 |
| CD24-BV510 (clone M1/69) | Biolegend | Cat# 101831 |
| CD3-biotin (clone KT3-1.1) | WEHI Antibody Facility | N/A |

| Antibody | Source | Identifier |
|---|---|---|
| CD38-BV786 (clone NIMR-5) | WEHI Antibody Facility | N/A |
| CD4-A680 | BD Biosciences | Cat# 550954 |
| CD40 (clone IC10) | WEHI Antibody Facility | N/A |
| CD43-BV786 (clone S7) | BD Biosciences | Cat #740857 |
| CD44-APC (clone IM7) | BD Biosciences | Cat# 103059 |
| CD62L-BUV737 (clone Mel14) | BD Biosciences | Cat# 565213 |
| CD69-PE-Cy7 (clone H1.2F3) | Biolegend | Cat #104511 |
| CD98-BV711 (clone H202-141) | BD Biosciences | Cat #745466 |
| CD98-BV421 (clone H202-141) | BD Biosciences | Cat #744831 |
| FAS-BUV395 (clone Jo2) | BD Biosciences | Cat# 740254 |
| Fc-γ-receptor block (anti-CD16/32, clone 2.4G2) | BD Biosciences | Cat #553141 |
| Gr-1-biotin (clone Rb6-8C5) | WEHI Antibody Facility | N/A |
| IgA-biotin (clone RMA-1) | BD Biosciences | Cat #407004 |
| IgD-biotin (clone 11-26c.2a) | WEHI Antibody Facility | N/A |
| IgD-PerCp-Cy5.5 (clone 11-26 c.2a) | BD Biosciences | Cat# 564273 |
| IgG1-BV421 (clone A85-1) | BD Biosciences | Cat# 562580 |
| IgM-A680 (clone 331.12) | WEHI Antibody Facility | N/A |
| IgM-PE-Cy7 (clone 11/14) | eBioscience | Cat #25-5790-82 |
| Igκ (clone 187.1) | WEHI Antibody Facility | N/A |
| Igλ (clone JC5) | WEHI Antibody Facility | N/A |
| Mac-1-biotin (clone M1/70) | WEHI Antibody Facility | N/A |
| Phospho$_{(Ser235/236)}$-S6-APC (clone D57.2.2E) | NEB | Cat #34411 |
| SDC4-PE (clone KY/8.2) | BD Biosciences | Cat# 550352 |
| Syk (pY348)-PE | BD Biosciences | Cat# 558529 |
| **Reagent** | **Source** | **Identifier** |
| Alhydrogel | InvivoGen | Cat# 21645-51-2 |
| Bovine serum albumin | Bovogen | Cat# BSAS1.0 |
| Cell Trace Violet (CTV) | Thermo Fisher Scientific | Cat #C34557 |

| Antibody | Source | Identifier |
|---|---|---|
| Chondroitin sulfate | Merck | Cat # 9082-07-9 |
| eBioscience™ Foxp3 / Transcription Factor Staining Buffer Set | Thermo Fisher Scientific | Cat# 00-5523-00 |
| EDTA-tetrasodium salt hydrate | Merck | Cat# E5391 |
| Fixable Viability Dye eFluor™ 780 | eBioscience | Cat# 65-0865-14 |
| HEL$^{WT}$-OVA$_{pep}$ and HEL$^{3x}$-OVA$_{pep}$ | In-house | N/A |
| Heparin | Hospira | Cat# 1089852 |
| HEPES solution | Sigma | Cat# H0887 |
| IL-21 (recombinant mouse) | Peprotech | Cat# 210-21-100 |
| IL-4 (recombinant mouse) | Biolegend | Cat# 574304 |
| IMDM | Gibco | Cat# 12440053 |
| NaHCO3 | Sigma | Cat# S5761 |
| NH4CL | Sigma | Cat# 254134 |
| Normal rat serum | Sigma | Cat# R9759 |
| PBS | Gibco | Cat# 14190-144 |
| Penicillin-Streptomycin | Sigma | Cat# P0781 |
| Polybrene | Merck | Cat# 20202471 |
| RPMI 1640 + GlutaMAX-I | Gibco | Cat# 61870-036 |
| Sodium pyruvate solution | Sigma | Cat# S8636 |
| Streptavidin-BUV496 | BD Biosciences | Cat# 564666 |
| Streptavidin-BV650 | BD Biosciences | Cat# 563855 |
| Streptavidin-BUV737 | BD Biosciences | Cat# 612775 |
| **Other** | **Source** | **Identifier** |
| BD LSRFortessa X-20 | BD | N/A |
| Coulter Counter | Beckman Coulter | N/A |
| Cytek Aurora | Cytek Biosciences | N/A |

## Mice

All mice were age and gender-matched mice on the C57BL/6 background. SW$_{HEL}$ mice were bred on a $Rag1^{-/-}$ background and contained transgenes for GFP and the heavy chain of the HyHEL10 monoclonal antibody specific to HEL as previously described (Phan et al, 2003; Quast et al, 2022). OT-II transgenic mice expressed a T-cell receptor (TCR) specific for ovalbumin residues 323-339 (Barnden et al, 1998). Rosa26-Cas9-GFP mice constitutively express GFP and Cas9 as previously described (Chu et al, 2016). All mice were bred and housed under specific pathogen-free conditions at the Monash Animal Research Platform (MARP) or at the Alfred Research Alliance (ARA) Monash Intensive Care Unit (MICU). Animal studies were approved by the A+ Research Alliance Animal Ethics Committee (applications: P8168 and P8211).

## Cell preparation and in vitro assays

Splenocytes were harvested by manual digestion of spleens through a 70 µM filter followed by resuspension in red blood cell removal buffer (156 mM $NH_4Cl$, 11.9 mM $NaHCO_3$, 0.01 mM tetrasodium EDTA). SW$_{HEL}$ B-cell cultures used $5 \times 10^5$ splenocytes per sample in media (RPMI, 10% fetal calf serum [FCS], 1 mM Na Pyruvate, 10 mM HEPES, 0.1 mg/mL Pen/Strep). For Fig. EV5A, WT B cells were enriched by staining splenocytes with biotinylated CD138, Gr-1, CD4, and CD8 followed by magnetic sorting using anti-biotin microbeads (Miltenyi Biotech) as previously described (Quast et al, 2022). Where applicable, cultures were supplemented with HEL$^{WT}$OVA$_{pep}$ (500 ng/mL), HEL$^{3x}$OVA$_{pep}$ (500 ng/mL), IL-4 (10 ng/mL), IL-21 (20 ng/mL), anti-CD40 (2 µg/mL), anti-Igκ (100 ng/mL) or anti-Igλ (100 ng/mL). Heparin was used at 2.5 (Figs. 4 and EV6D,E) or 250 IU/ml (Fig. EV6A–C), HS, CS, and polybrene at 5 µg/ml. For proliferation assays, splenocytes were stained for 10 min with 5 µM Cell Trace Violet (Thermo Fisher). To remove HS, splenocytes were treated with heparinase II (New England Biolabs; 16 IU/mL) for 3 h, 37 °C before centrifugation and resuspension in media for culture.

## In vitro B-cell differentiation

In vitro B-cell differentiation was done as previously described (Holl et al, 2010). Frozen BM from $Sdc1$ or $Sdc4$-targeted mice was thawed and resuspended in 10 ml pre-heated (37 °C) IMDM media. Samples were centrifuged (4 °C; $400 \times g$; 5 min), resuspended in 10 ml supplemented IMDM media (10% FBS, 55 µM 2-ME, penicillin (10 U/ml) and streptomycin (10 µg/ml)) and incubated in culture dishes for 15 min at 37 °C in a humidified cell culture incubator. Non-adherent cells were collected, centrifuged, resuspended in 1 ml red cell removal buffer, and washed. Cells were resuspended in 1 ml IMDM and counted. In total, 250 µl of this cell suspension were analyzed by flow cytometry (day 0), and the remainder ($0.75 - 1.5 \times 10^7$ cells) was plated in 8 ml supplemented IMDM media containing 4% cell culture supernatant of an IL-7 expressing cell line (provided by Stephen L. Nutt (Greig et al, 2010)) and incubated in a humidified incubator. On day 4, non-adherent cells were aspirated, centrifuged, resuspended in 1 ml supplemented IMDM, counted, and 330 µl removed for analysis by flow cytometry (day 4). The remainder was resuspended in supplemented IMDM containing IL-7 and re-plated for 4 more days, after which cells were counted and analyzed by flow cytometry (day 8).

## Flow cytometry

All samples were stained for surface molecules with antibodies in 2% BSA in PBS supplemented with 1% normal rat serum and Fc-γ-receptor block (anti-CD16/32, clone 2.4G2). Intracellular staining was conducted using the eBioscience Foxp3/Transcription Factor Staining Buffer Set (Thermo Fisher) as per the manufacturer's instructions. Samples were washed in 2% BSA and passed through a 70 µM filter prior to acquisition. Flow cytometry data was acquired using an BD LSRFortessa X-20 or a Cytek Aurora flow cytometer and analyzed with FlowJo (BD).

## Adoptive transfer and immunization

Adoptive transfer and immunization were performed as previously described (Dvorscek et al, 2022; Dvorscek et al, 2024). Briefly, the

frequency of B cells in SW$_{HEL}$ and CD4$^+$ T cells OT-II splenocytes was determined by flow cytometry. $1 \times 10^5$ SW$_{HEL}$ B cells and with $5 \times 10^4$ OT-II CD4$^+$ T cells were intravenously injected into recipient C57Bl/6 WT mice using 27 G needles. For immunization, 50 µg HEL$^{WT}$OVA$_{pep}$ was mixed with 45 µL Alhydrogel (Invivo-Gen), incubated for 30 min at room temperature, suspended to a final volume of 200 µL in PBS, and administered i.p. within 2 h of cell transfer.

## Lentiviral vector production

Guide RNA sequences targeting murine *Sdc1* (CTTCCACTTG-GAAGGACGTG) and *Sdc4* (CATGTCATCCCCCACGTCGG) were identified using Alt-R CRISPR–Cas9 guide RNA tool (Integrated DNA Technologies). Oligos containing complementary sgRNA sequences and overhangs (Forward 5': CACC, Reverse 5': AAAC) were annealed and cloned into BbsI-digested transfer plasmid pKLV-U6gRNA(BbsI)-PGKpuro2ABFP (Addgene). Lentiviral particles were generated as previously described (Aubrey et al, 2015; Kueh and Herold, 2016). Briefly, HEK 293T cells cultured in DMEM (10% FCS, 25 mM HEPES) were transiently transfected in 10 cm dishes with 7.5 µg of transfer plasmid in conjunction with packaging constructs pMDL (3.8 µg), pRSV-rev (1.9 µg) and pVSV-G (2.3 µg). Supernatants containing lentiviral vectors were collected 48–72 h post-transfection and stored at −80 °C.

## CRISPR–Cas9 genome editing and BM reconstitution

BM from Cas9 transgenic mice was isolated by mechanical disruption of femora and tibiae followed by resuspension in Opti-MEM media (Thermo Fisher) and passage through a 70 µM filter. BM was cultured in Opti-MEM supplemented with 10% FCS, 1 mM Na Pyruvate, 10 mM HEPES, 0.1 mg/mL Pen/Strep, 10 ng/mL IL-6 (Merck, Cat. SRP333), 100 ng/mL stem cell factor (SCF; Merck, Cat. S9915), 50 ng/mL thrombopoietin (TPO; Merck, Cat. SRP3236) and 10 ng/mL FMS-like tyrosine kinase 3 ligand (Flt3l; Merck, Cat. SRP3198). Following 24 h in culture, BM was transduced with lentiviral vectors encoding sgRNA to either *Sdc1* or *Sdc4* and a BFP reporter. Cultured BM was spinfected with lentiviral supernatants supplemented with 10 µg/mL polybrene for 1.5 h, 2200 rpm, 32 °C on plates pre-coated with retronectin (Takara Bio). The following day, BM was washed, resuspended in PBS, and supplemented with 20% freshly prepared BM from a RAG-1$^{-/-}$ mouse (to ensure reconstitution of the hematopoietic system). WT recipient mice were irradiated (2 doses, 5.5 Gy, 3 h apart), and $1.3 \times 10^6$ BM cells were injected i.v. Reconstituted mice received neomycin (1.622 mg/mL in drinking water) for 2 weeks and were immunized i.p. with alum-adjuvanted NP-KLH (100 µg) 8 weeks post reconstitution as previously described (Zotos et al, 2021). Spleen and BM were harvested 1-week post-immunization.

## Statistics

All data were analyzed in Graph Pad Prism with statistical tests specified for each figure in the corresponding figure legend. No blinding was done. One mouse did not respond to NP-KLH immunization and was excluded from analysis (Fig. 1F). Data with $P \leq 0.05$ were considered statistically significant.

## Data availability

No primary datasets have been generated and deposited.

The source data of this paper are collected in the following database record: biostudies:S-SCDT-10_1038-S44319-025-00432-6.

## Peer review information

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

## Acknowledgements

The authors thank Robert Brink for providing SW$_{HEL}$ and William Heath for OT-II mice, the ARAFlowCore for assistance with flow cytometry, CSIRO for HEL-OVA$_{pep}$ protein production, Alfred Research Alliance (ARA) Monash Intensive Care Unit and Monash Animal Research Platform for animal husbandry. The synopsis image was created in BioRender.com (license agreements with IQ; https://BioRender.com/d73p478The). ARD was supported by a Monash University Research Training Program stipend and a Career Advancement Award provided by the Australian and New Zealand Society for Immunology. ZD and MJR were supported by a Monash University Future Leader Fellowship and MJR by a National Health and Medical Research Council (NHMRC) Australia Ideas Grant (APP1185294, awarded to MJR and IQ). JM was supported by a Monash University Research Training Program stipend. DMT was funded by a NHMRC Australia Investigator Award (APP1175411). IQ was supported by a Peter Doherty Early Career Fellowship (APP1145136) provided by NHMRC Australia.

## Author contributions

**Craig I McKenzie**: Conceptualization; Data curation; Formal analysis; Investigation; Visualization; Methodology; Writing—original draft; Writing—review and editing. **Alexandra R Dvorscek**: Data curation; Formal analysis; Investigation; Visualization; Methodology; Writing—review and editing. **Zhoujie Ding**: Investigation; Writing—review and editing. **Marcus J Robinson**: Funding acquisition; Methodology; Writing—review and editing. **Kristy O'Donnell**: Investigation. **Catherine Pitt**: Investigation. **Daniel T Ferguson**: Methodology. **Jesse Mulder**: Investigation. **Marco J Herold**: Resources; Methodology. **David M Tarlinton**: Funding acquisition; Methodology; Writing—review and editing. **Isaak Quast**: Conceptualization; Formal analysis; Supervision; Funding acquisition; Validation; Methodology; Writing—original draft; Writing—review and editing.

Source data underlying figure panels in this paper may have individual authorship assigned. Where available, figure panel/source data authorship is listed in the following database record: biostudies:S-SCDT-10_1038-S44319-025-00432-6.

## Disclosure and competing interests statement

The authors declare no competing interests.

# Expanded View Figures

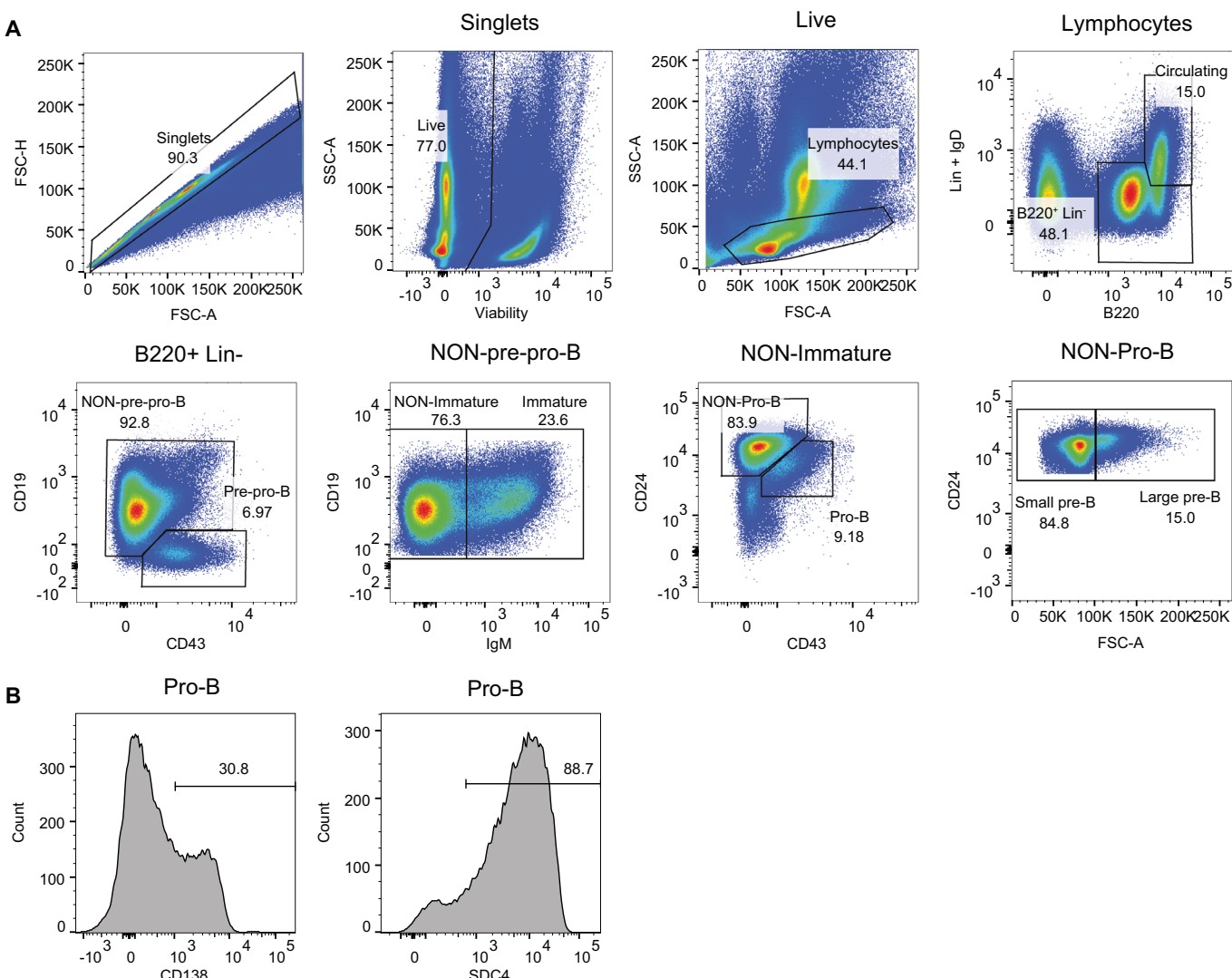

**Figure EV1. Flow cytometric gating strategy for the identification of B-cell subsets undergoing maturation in the BM.**

(A) Representative flow cytometric plots for the identification of the indicated stages of B-cell development in BM. Lin (Mac-1, Gr-1, CD3). (B) Representative histograms of CD138 and SDC4 expression on pro-B cells.

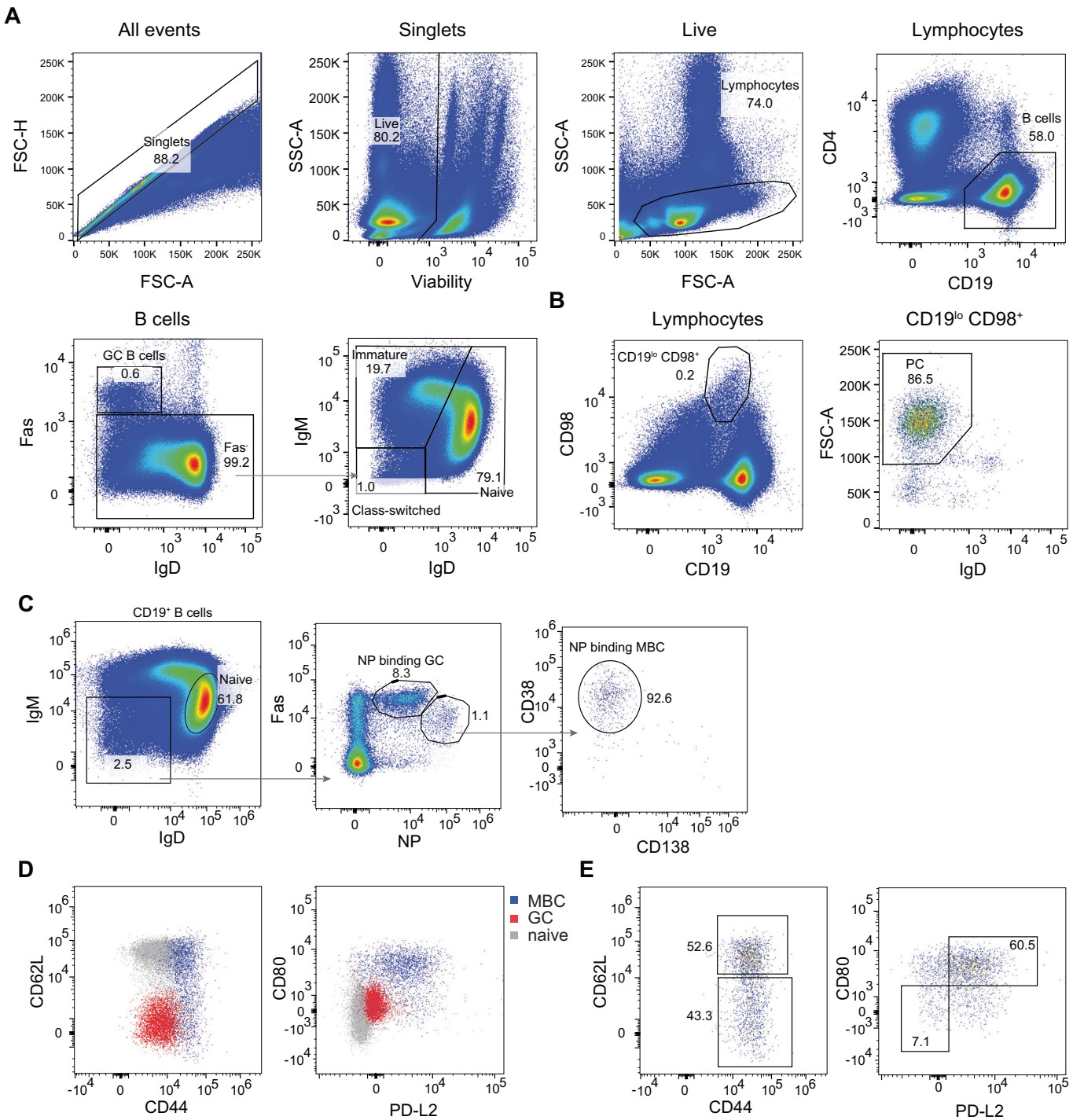

**Figure EV2. Flow cytometric gating strategy for the identification of mature B cell subsets in the spleen.**

(A, B) Representative flow cytometric plots for the identification of (A) B cells and B-cell subsets and (B) PC. (C–E) Analysis of antigen (NP)-specific germinal center (GC) and MBC 3 weeks post NP-KLH immunization. (C) MBC gating strategy, (D) color-coded overlay of indicated population showing expression of cell surface molecules with MBC subsets. (E) MBC subset gating.

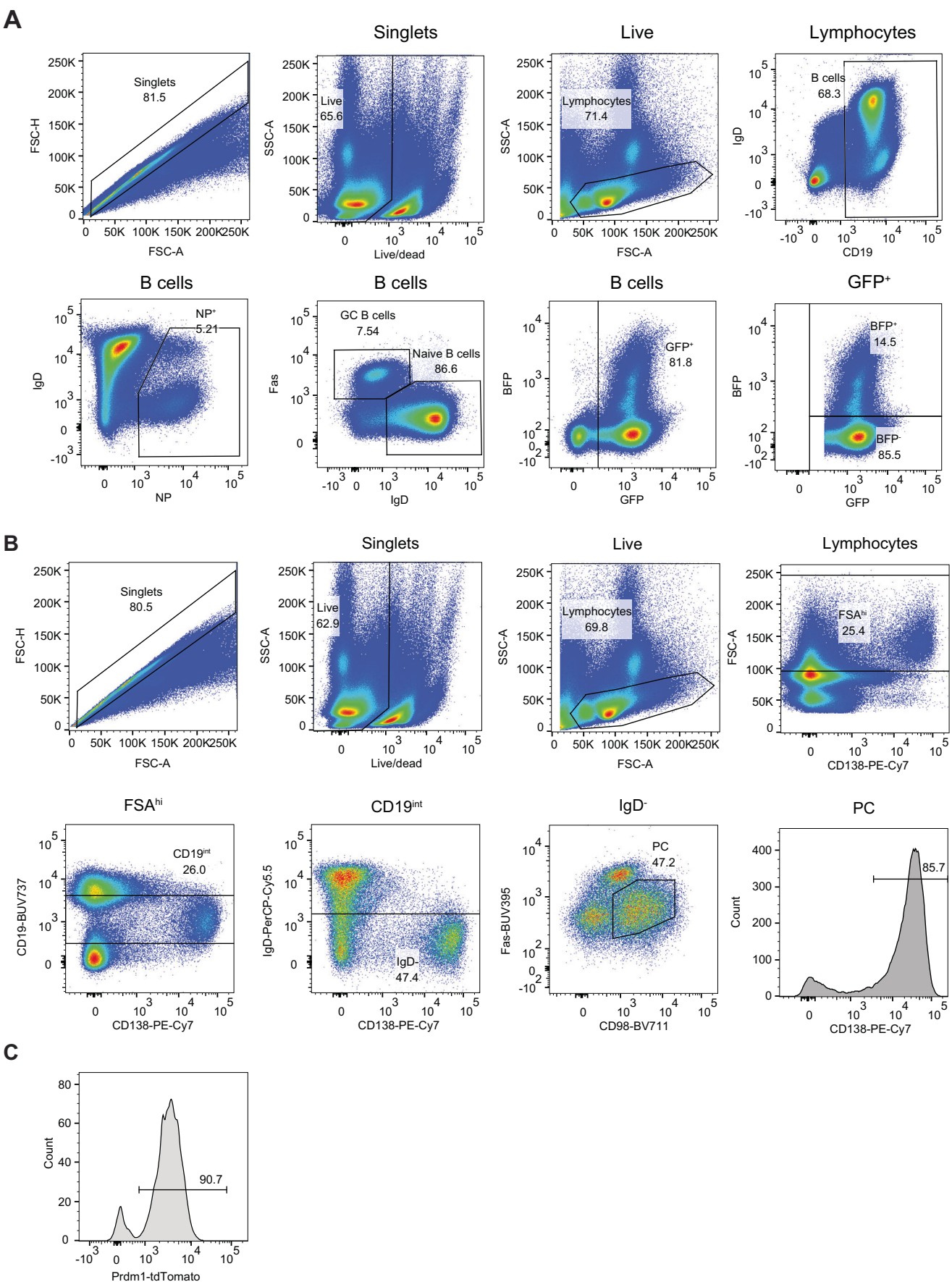

**Figure EV3. Flow cytometric gating strategy for the identification of B-cell subsets from the spleen of reconstituted mice.**

(A) Representative flow cytometry plots showing sequential gating steps to identify mature B-cell subsets and NP-specific B cells in the spleen of mice reconstituted with lentivirally transduced BM for CRISPR–Cas9 genome editing. (B) Representative flow cytometry plots for the identification of PC from the spleen of reconstituted mice. (C) CD138 independent PC gating strategy applied to splenocytes from mice expressing tdTomato under the control of the *Prdm1* locus.

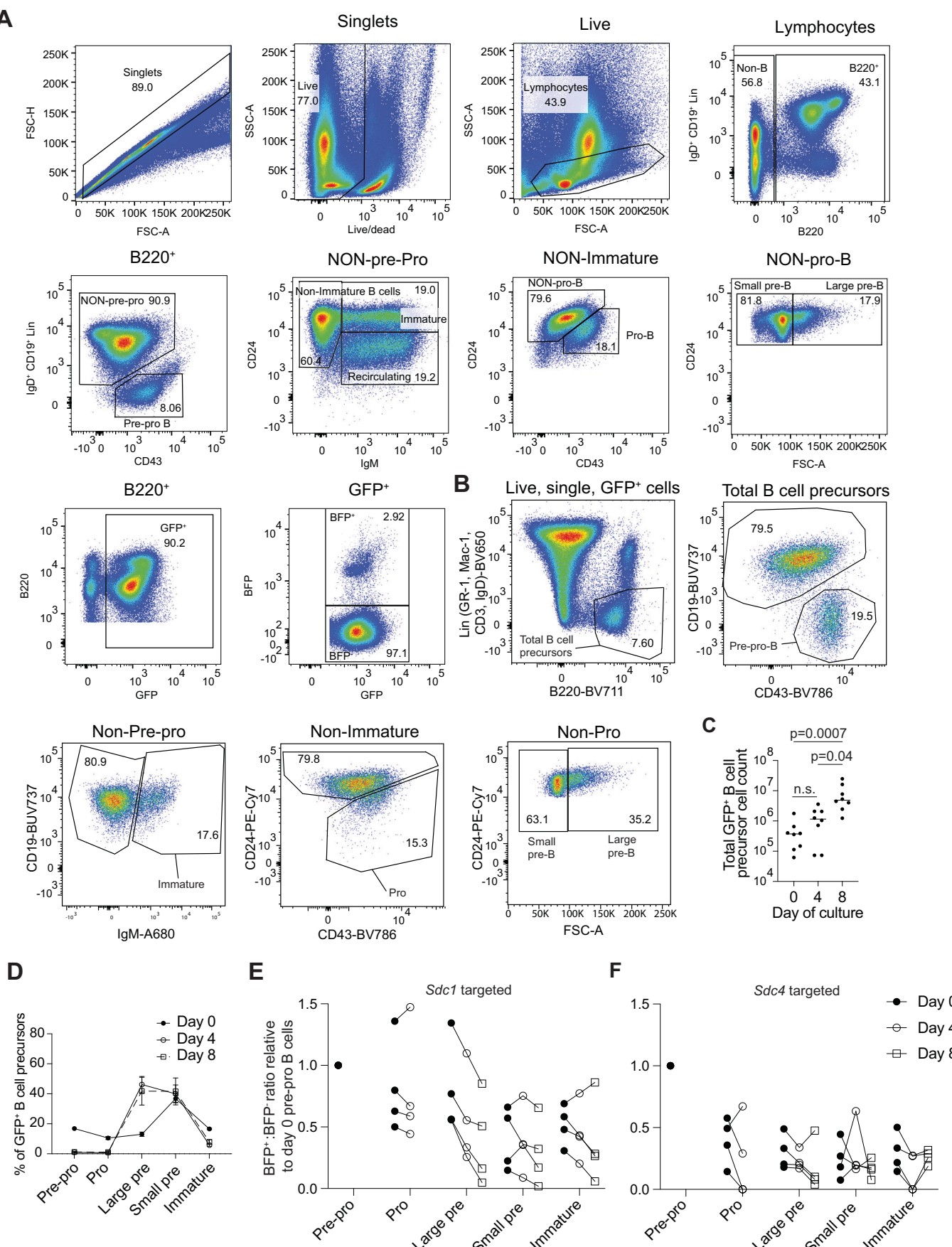

◀ **Figure EV4. Flow cytometric gating strategy for the identification of developing B cell subsets in the BM of mice reconstituted with lentiviral transduced Cas9 transgenic BM.**

(A) Representative flow cytometry plots for the identification of the indicated stages of B-cell development in BM. Lin (Mac-1, Gr-1, CD3). (B–F) In vitro IL-7 mediated B-cell differentiation. gating strategy (B), total B-cell precursor cell lumbers (GFP$^+$B220$^+$ Lin$^-$ cells) (C) and precursor subset composition (D) over time. (E) Proportional representation of *Sdc1* gene-targeted (GFP$^+$BFP$^+$) cell among CAS9$^{tg}$ BM-derived (GFP$^+$) B-cell precursors relative to Pre-pro-B cells at the start of culture (day 0). (F) Analysis as in (E) but for Sdc4-targeted cells. Statistical analysis in (C) by Kruskal–Wallis test with Dunn's multiple comparisons test. Data in (D–F) are pooled from two experiments. (D) shows means ($n = 8$) and SEM. Lines in (E) and (F) connect individual mice ($n = 4$). Some data points in pre-pro and pro-B cells are absent due to very low cell numbers at the respective time points. Source data are available online for this figure.

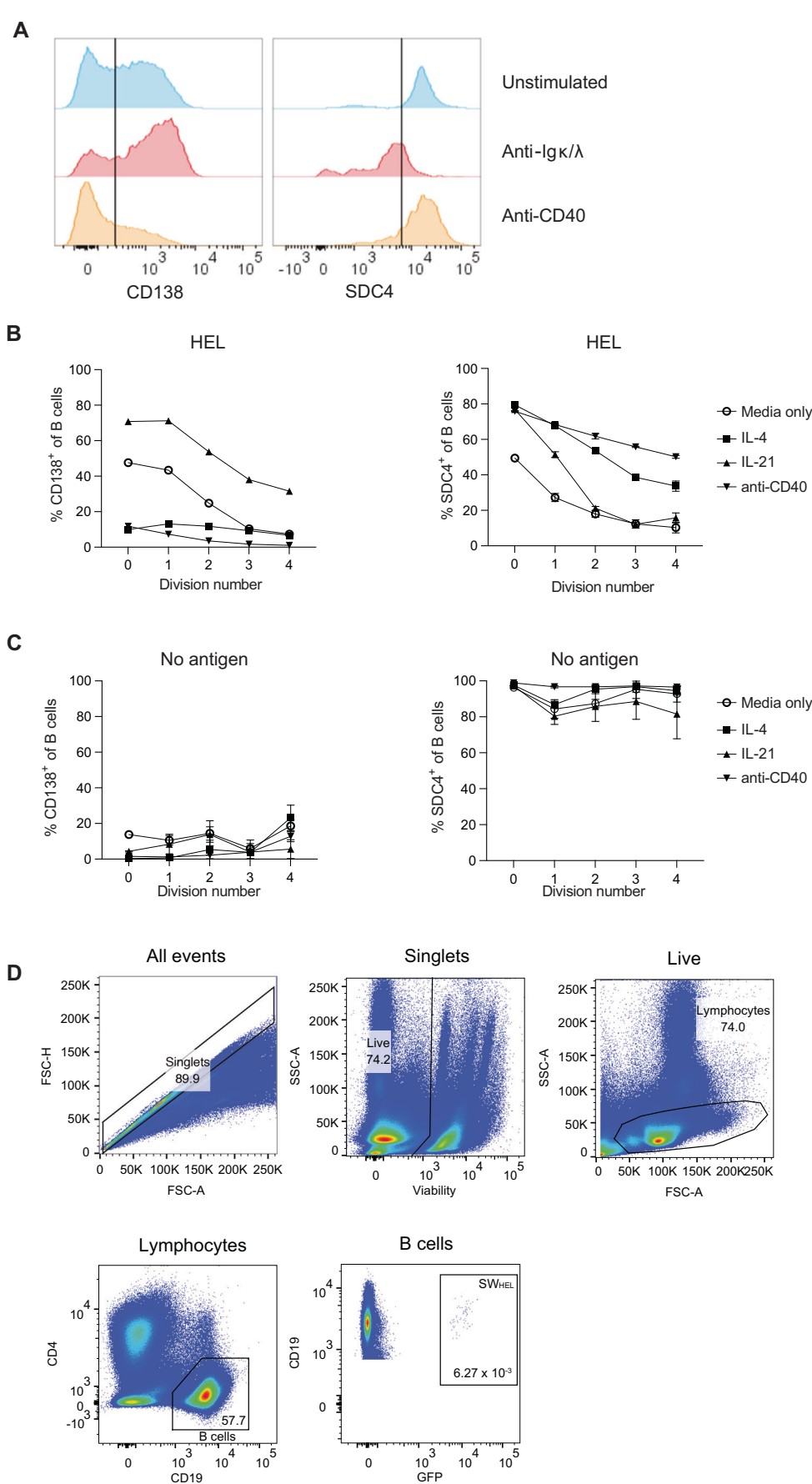

**Figure EV5. Expression of CD138 and SDC4 on BCR-stimulated B cells decreases with consecutive divisions.**

(A) CD138 and SDC4 expression on B cells stimulated for 3 days in culture with anti-Igκ and anti-Igλ or anti-CD40 antibody. (B, C) Expression of CD138 and SDC4 on RAG-1$^{-/-}$ SW$_{HEL}$ B cells stimulated with (B) or without (C), HEL and IL-21, IL-4 or anti-CD40 (additional data for Fig. 3) showing mean and SD of technical triplicates, representative of 2 experiments. (D) Gating strategy for the identification of RAG-1$^{-/-}$ SW$_{HEL}$ B cells 3 days post-transfer in the spleen of HEL$^{WT}$-OVA$_{pep}$ immunized wild-type recipients. Source data are available online for this figure.

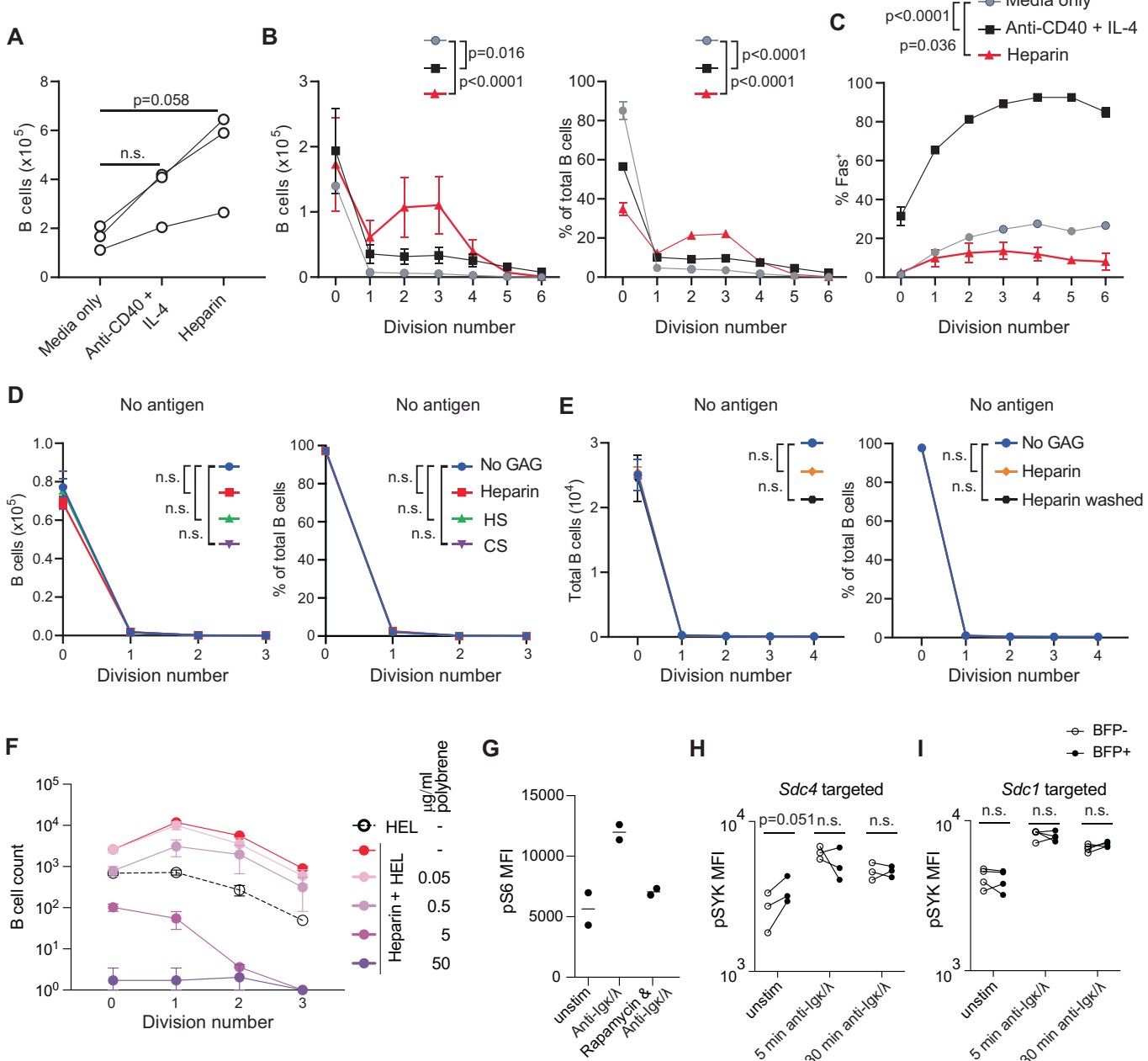

**Figure EV6. Heparin at a high concentration induces B-cell proliferation without FAS upregulation.**

(A) Total number of WT B cells, (B) number and proportion of B cells per division and (C) proportion of FAS+ B cells per division after 3 days of culture with high-dose heparin (250 IU/mL) or anti-CD40 and IL-4. (D, E) Number and proportion of RAG-1−/− SW_HEL B cells per division after 3 days of culture without HEL antigen and (D) with heparin (2.5 IU/mL), HS and CS or (E) after short-term pre-incubation with heparin (2.5 IU/mL) followed by washing and re-culture for 3 days (additional data for Fig. 4). (F) RAG-1−/− SW_HEL B cells count per CTV division peak after 3 days cell culture in the presence of HEL or HEL + heparin (2.5 IU/mL) with or without polybrene. (G) WT B cells (n = 2) were incubated for 30 min with anti-Igκ and anti-Igλ in the presence or absence of rapamycin and S6 phosphorylation (pS6) analyzed by flow cytometry. (H, I) Syk phosphorylation (pSyk) in B cells from BM-reconstituted mice (as shown in Fig. 3) comparing non-targeted to Sdc4 (H) or Sdc1 (I) sgRNA-targeted B cells. (A–E) are representative of 2 experiments and show mean of technical triplicates (n = 3) with error bars indicating standard deviation. Data in F are from one experiment, assayed in duplicates showing median and range. H and I show data from 3–4 mice (cells from each mouse were assayed in duplicates and means are shown) and statistical analysis by paired t test. Source data are available online for this figure.

