## [Peer Review File · EMBO Reports]

Syndecans and glycosaminoglycans influence B cell development and activation

Craig McKenzie, Alexandra Dvorscek, Zhoujie Ding, Marcus Robinson, Kristy O'Donnell, Catherine Pitt, Daniel Ferguson, Jesse Mulder, Marco Herold, David Tarlinton, and Isaak Quast

Corresponding author(s): Isaak Quast (isaak.quast@monash.edu) , Isaak Quast (isaak.quast@monash.edu), Craig McKenzie (craig.mckenzie@monash.edu)

Review Timeline:

Submission Date:	19th Jun 24
Editorial Decision:	15th Jul 24
Revision Received:	5th Feb 25
Editorial Decision:	27th Feb 25
Revision Received:	7th Mar 25
Accepted:	12th Mar 25

Editor: Achim Breiling

Transaction Report:

Dear Dr. Quast,

Thank you for the transfer of your manuscript to EMBO reports. I have now received the reports from the three referees that were asked to evaluate your study, which can be found at the end of this email.

As you will see, the referees have several comments, concerns, and suggestions, indicating that a major revision of the manuscript is necessary to allow publication of the study in EMBO reports. As the reports are below, and all the concerns need to be addressed, I will not detail them further here.

Given the constructive referee comments, I would like to invite you to revise your manuscript with the understanding that the concerns of the referees must be addressed in the revised manuscript and in a detailed point-by-point response. Acceptance of your manuscript will depend on a positive outcome of a second round of review. It is EMBO reports policy to allow a single round of revision only and acceptance of the manuscript will therefore depend on the completeness of your responses included in the next, final version of the manuscript.

- 1) a .docx formatted version of the final manuscript text (including legends for main figures, EV figures and tables), but without the figures included. Figure legends should be compiled at the end of the manuscript text.
- 2) individual production quality figure files as .eps, .tif, .jpg (one file per figure), of main figures and EV figures. Please upload these as separate, individual files upon re-submission.

- 4) a complete author checklist, which you can download from our author guidelines (<https://www.embopress.org/page/journal/14693178/authorguide>). Please insert page numbers in the checklist to indicate where the requested information can be found in the manuscript. The completed author checklist will also be part of the RPF.

- 5) that primary datasets produced in this study (e.g. RNA-seq, ChIP-seq, structural and array data) are deposited in an

appropriate public database. If no primary datasets have been deposited, please also state this in a dedicated section (e.g. 'No primary datasets have been generated and deposited'), see below.

The accession numbers and database should be listed in a formal "Data Availability" section (placed after Materials & Methods) that follows the model below. This is now mandatory (like the COI statement). Please note that the Data Availability Section is restricted to new primary data that are part of this study. This section is mandatory. As indicated above, if no primary datasets have been deposited, please state this in this section

Data availability

8) Regarding data quantification and statistics, please make sure that the number "n" for how many independent experiments were performed, their nature (biological versus technical replicates), the bars and error bars (e.g. SEM, SD) and the test used to calculate p-values is indicated in the respective figure legends (also for EV figures and all those in an Appendix). Please also check that all the p-values are explained in the legend, and that these fit to those shown in the figure. Please provide statistical testing where applicable. Please avoid the phrase 'independent experiment', but clearly state if these were biological or technical replicates. Please also indicate (e.g. with n.s.) if testing was performed, but the differences are not significant. In case n=2, please show the data as separate datapoints without error bars and statistics. See also: <http://www.embopress.org/page/journal/14693178/authorguide#statisticalanalysis>

9) Please add scale bars of similar style and thickness to microscopic images, using clearly visible black or white bars (depending on the background). Please place these in the lower right corner of the images themselves. Please do not write on or near the bars in the image but define the size in the respective figure legend.

10) Please also note our reference format:

12) We now use CRedit to specify the contributions of each author in the journal submission system. CRedit replaces the author contribution section. Please use the free text box to provide more detailed descriptions and do NOT provide your final manuscript text file with an author contributions section. See also our guide to authors: <https://www.embopress.org/page/journal/14693178/authorguide#authorshipguidelines>

13) All Materials and Methods need to be described in the main text using our 'Structured Methods' format, which is required for

all research articles. According to this format, the Materials and Methods section should include a Reagents and Tools Table (listing key reagents, experimental models, software, and relevant equipment and including their sources and relevant identifiers), uploaded as separate file, followed by a Methods and Protocols section in which we encourage the authors to describe their methods using a step-by-step protocol format with bullet points, to facilitate the adoption of the methodologies across labs. More information on how to adhere to this format as well as downloadable templates (.doc) for the Reagents and Tools Table can be found in our author guidelines (section 'Structured Methods'):

14) Please order the manuscript sections like this, using these names:

Title page - Abstract - Keywords - Introduction - Results - Discussion - Methods - Data availability section - Acknowledgements - Disclosure and Competing Interests Statement - References - Figure legends - Expanded View Figure legends

Finally, please note that all corresponding authors are required to supply an ORCID ID for their name upon submission of a revised manuscript. Please do that for the co-corresponding author Craig McKenzie. Please find instructions on how to link the ORCID ID to the account in our manuscript tracking system in our Author guidelines: <http://www.embopress.org/page/journal/14693178/authorguide#authorshipguidelines>

I look forward to seeing a revised form of your manuscript when it is ready.

Yours sincerely,

Referee #1:

McKenzie et al describe "The role of syndecans in B cell development, activation and plasma cell formation".

Plasma cells, by secreting antibodies, are obviously central to humoral immunity. Plasma cells are characterized by high expression of CD138, but the role of this glycosaminoglycan-containing protein besides serving as a marker of plasma cells for immunologists is incompletely understood.

Here, the authors study the expression and function of CD138 (Syndecan-1) and of another syndecan expressed in B cells (syndecan-4) using state of the art techniques in at least 10 B cell stages between pre-pro B cells and plasma cells. It is noteworthy that their study stopped at plasma cell generation and was not designed to study how PCs are maintained in the long-term. The work is divided in 4 main blocks.

In a first part, the detail surface expression of CD138 and Syndecan4 is studied. In a second part, authors monitor, using bone marrow chimera, how stem cells knock-out for Sdc1 (CD138) or Sdc4 compare to WT to reconstitute B cell populations. Results show that neither Sdc1 nor Sdc4 are essential for B cell development. Indeed plasma cells can be produced and are alive in the absence of one or the other. In both ko, B cell development is impaired about 50% usually at the stage where these molecules are most highly expressed. This indicates a positive but non-essential function of Sdc1 and Sdc4 during development.

In a third part, regulation of SDC1 and SDC4 expression was studied in naïve B cells that express SDC4 but not SDC1. Direct stimulation of the BCR permanently decreased SDC4 expression and transiently increased SDC1 expression. Other stimuli tested (IL4, IL21, anti-CD40 [all produced by or mimicking T helper cells]) had no effect alone, but tended to maintain the status quo of SDC1 and 4 expression upon BCR stimulation (and therefore go against the BCR effect on SDC1 and 4, but without preventing proliferation). The exception was IL21 that further promoted transient BCR-mediated CD138 expression. Consistent

results of SDC4 downregulation and SDC1 expression were observed when B cells were stimulated with antigen and T helper cells in vivo.

In the fourth part, authors studied the effect of adding moderate or excess amounts of soluble glycosaminoglycans to naïve B cells {plus minus} cognate antigen. In a "reverse" experiment, they digested glycosaminoglycans on B cells with heparinase. Although effects were present and sometimes spectacular (addition of glycosaminoglycans increased B cell number up to almost 10-fold; addition of glycosaminoglycans reduced activation markers CD69 or Fas or loss of SDC4 induced by BCR stimulation; removal of cell-bound glycosaminoglycans with heparitinase increases CD69 expression post BCR stimulation), I found it difficult to understand how authors conclude that glycosaminoglycans and in particular SDC4 have an inhibitory function on B cells, and to figure out by which mechanism this could be.

In summary, the experiments are very well performed, figures are of high quality, the introduction is to the point and some important conclusions are fully supported by data (SDC4 and CD138 are not essential for B cell development and production of plasma cells). The conclusion that syndecans (at least SCD4) are inhibitory for naïve B cells is probably supported by data, but this could be better explained. Often, conclusions are written in vague (or cautious) terms that do not help the reader understand what is going on (eg title: "The role of syndecans in B cell development, activation and plasma cell formation." OK, but what is this role? Eg abstract: "Collectively, our results reveal SDCs as regulators of the B cell response to antigen, with GAGs modulating B cell activation and survival." The terms "regulator" and "modulating" gives no indication on the direction or amplitude of the modulation. Etc...

Major point 1: The authors should clarify their descriptions of Fig 5 and Fig 6 investigating the role of GAGs in naïve B cell activation. For example, Fig 5A shows that heparin has no effect on the number of naïve B cells in the absence of antigen stimulation, but that in the presence of the antigen (HEL), there are about 10-fold more B cells in the presence of heparin than in its absence. It is very difficult for the reviewer to reconcile this result with the final conclusion of the authors that GAGs have an inhibitory role on B cells. One possibility is that under these conditions, the high affinity ligand induces death of the stimulated cells (AICD) which would explain the strongly reduced cell number of the condition "no GAGs" in the presence of antigen compared to the absence of antigen. AICD could be due to upregulation of FAS which is counteracted to some extent by the presence of heparin. As FAS is an activation marker, then heparin would negatively regulate activation, keep cells under the threshold of AICD and prevent cells from dying. If this is the case, it could be explained better, eg in a wrap up sentence. But then it is unclear why in Fig 5C that tests the effect of short-term naïve B cells exposure to heparin there is no difference in B cell number in the "No GAG" condition minus or plus antigen, and that this is also not seen in Fig 5F in No GAG minus or plus antigen. Fig 6A however resembles Fig 5A in that naïve B cell number is drastically reduced upon stimulation with antigen (probably by AICD). This defect is attenuated by heparin, but never to the extent seen in Fig 5A. Experimental variability? Different experimental conditions?

Also, it would be useful for the reader if authors could integrate results of Fig 4 with their conclusion that syndecans have an inhibitory role on B cells. For example, is it conceivable that T cell help would prevent premature differentiation of cycling B cells in antibody secreting cells and plasma cells by among others, preventing CD138 expression and preventing SDC4 down-regulation, with the aim of achieving a better cell amplification before terminal differentiation?

Minor point 2: In the experiment of Figure 4, naïve B cells not only express anti-HEL BCR, but also come from a Rag1-ko background. It would be helpful in all figures using B cells with anti-HEL BCR to mention if they are also Rag1-ko or not.

Minor point 3: In Fig 4A, the CTV profile indicates that CD138 that is not expressed in naïve B cells was induced by HEL even before division, and then fully lost again after 3 divisions. This profile looks exactly like the one of Fig 4B with SDC4, except that here SDC4 was already present and is just lost upon HEL stimulation within 3 divisions. This could be better explained in the text. What justifies this very transient expression is a mystery for me. It cannot be directly linked to a differentiation to PC. Are Fig 4A,B comparable in terms of cells and HEL activation to Fig 5 and 6? If yes and if CD69 and/or FAS staining were included in these experiments (of Fig 4A,B), and if SDC4 indeed prevents activation, one would expect to see CD69 go up as cells divide in the Media only + HEL condition, but not in the anti-CD40 + HEL condition. If this data is available, please show the results (and possibly revise the conclusion of the inhibitory role of SDCs as necessary). If these stainings were not included, I let authors decide whether they would like to do it, but I do not insist that the experiment should be done.

Minor point 4: It is unclear how addition of heparin, or digestion with heparinase would interfere with SDC4 to exert its effect on B cells. A) GAGs of SDC4 inhibit B cell activation directly (e.g. by binding the BCR). Adding heparin allows to inhibit B cells even more (e.g. by binding more BCR). The digestion with heparinase removes GAGs and less partner is inhibited. B) GAGs of SDC4 bind a positively charged partner (APRIL, IL6, IL21?) that is displaced by an excess of heparin and thus stops activating B cells. What happens if naïve B cells are digested with heparinase, then washed, then stimulated with heparin? If heparin can bind and inhibit the same target as SDC4, this should cancel the action of heparitinase (I do not insist to have this experiment done).

Referee #2:

Syndecans can play important roles as modulators of cytokine and growth factor responses and can mediate attachment of cells. They can also act as co-receptors for GPRs. This report aims at addressing the abundance and function of two Syndecans, Sdc1 and 4, in B cells. To this end, bone marrow of Cas9tg mice was infected with lentiviral vectors encoding specific sgRNAs and a BFP reporter. Infected BM along with RAG1^{-/-} BM was transplanted into irradiated WT mice and B cell development and activation were analyzed. While CD138 peaks at pre B cells and in plasma cells, Sdc4 becomes up-regulated in pre and immature B cells and down-regulated upon B cell activation as well as in memory B cells, however, its expression differs among memory B cell subsets.

Functionally, deletion of both CD138 and Sdc4 disfavoured early B cell development. While Sdc4 B cells reached control frequency, the disadvantage of CD138 targeted B cells remained until the immature B cell stage. Immunization of recipient mice with NP-KLH did not reveal any effects of CD138 or Sdc4 deletion on the frequency of deleted germinal center B cells or plasma cells.

As to B cell activation, the SwHEL system was used in combination with IL-4, IL-21 and anti CD40. While IL-21 promoted HEL induced CD138 upregulation, IL-4 and CD40 suppressed it. The reverse was observed for Sdc4.

To address functional effects by putative Sdc ligands such as Polybrene and Heparin, activated B cells were co-cultured with either substance or treated with Heparinase. Heparin acted as B cell mitogen and is discussed as putative TLR4 ligand, hence being not so much relevant for human B cells in that respect. On the other hand, Heparin (HEL stimulation) suppressed AICD and CD69 expression. Heparinase (3xHEL) had no effect on AICD but increased CD69 expression. Sdc4 limited pS6 in resting and anti Ig activated B cells.

In summary, the presented data serve as a good starting point to pinpoint the role of CD138 and Sdc4 in B cells.

1. There are only few functional data regarding CD138 or Sdc4 knock-out, but both appear to affect early B cell development, which is interesting. What is the mechanism behind the disadvantage of CD138 and Sdc4 deficient pre/pro/pre B cells? PS6 needs to be downregulated to ensure proper small pre B cell development and Sdc4 appears to modulate pS6 in activated B cells. Given that the authors outstanding expertise in BM and stem cell cultures, would it not be feasible to set up stem cell / IL-7 cultures with CD138 or Sdc4 targeted stem cells and analyze pre B cell differentiation in vitro, defining proliferation/apoptosis/signaling/small pre B cell differentiation (using CD19/B220/CD2/IgM staining)? Pre BCR ligands should be discussed. It is also possible that in vitro cultures show no effect as the potential Sdc ligand may only be present in the bone marrow. However, I believe that the role of Sdcs in B cell development should be explored more here.

2. Was the pS6 in activated B cells normalized to total S6? Can pS6 be blocked by Rapamycin? I think the conclusion that Sdc4 is a negative regulator of BCR signaling is a bit premature. Is BCR induced pSyk/PLCg2/Ca2+ affected or are the observed effects downstream of proximal BCR signaling?

3. Conceptually, it would be important to assess the effects of Heparin and Heparinase also on Sdc4KO B cells.

4. What happens if B cells are incubated with both Polybrene and Heparin?

5. Why did the authors use HEL in Figure 6 A and 3 x HEL in Figure 6B? This is a bit misleading and hard to grasp.

Referee #3:

The manuscript of McKenzie and colleagues provides a comprehensive overview on expression of the Glycosaminoglycan (GAG) containing cell surface proteins SDC1 and SDC4 during differentiation and activation of B lymphocytes. Based on this overview, the impact of genetic ablation of SDC1 or SDC4 in mice, and the impact of soluble GAGs on human and murine B cell activation is analysed, with the key claims that SDC4 is a negative modulator of B cell activation and SDC1 (CD138) is not relevant for differentiation of activated B cells into plasma cells. As such, the manuscript adds important information on the regulation of B cell differentiation and activation by GAGs, of interest to the readership of EMBO Reports, in light of the central role of B cells in providing humoral immunity and autoimmunity. However, some issues have to be clarified upfront. The major drawback seems to be that the entire readout of expression is based on cytometric data, and not adjusted to (single cell) transcriptome data, for staging of B cell differentiation. This would clearly put the manuscript on a different level.

The descriptive part of the manuscript, i.e. Figs 1 and 2, and EVFigs 1 and 2, is rather clear, as far as the expression pattern of SDC1 and SDC4 in the general steps of B cell differentiation is concerned. However, when analysing the bimodal expression of SDC1 and 4 in experienced B cells, and using CD62L and CD44 to discriminate between recent GC emigrant MBC (CD62L+CD44-) and established MBC (CD62L+CD44+), the data shown in EVFig 2C show that the mice analysed did not have any CD62L+CD44+ MBC, for whatever reason, maybe the mice were just too young and immunologically inexperienced. However, in Fig. 2A bimodal distributions are shown for CD62L-CD44- MBC and CD44-CD62L+ MBC, not for CD44+CD62L+, but then in Fig. 2B, frequencies are provided for these MBC. These data are not convincing. The differential expression of SDC1 and 4 on MBC remains enigmatic, the data provided are inconsistent, and mice with significant numbers of advanced MBC

should be analysed.

When it comes to the controversial point, and as the authors claim key statement, of whether SDC1 (CD138) does impact on plasma cell differentiation, as suggested by an earlier publication (McCarron et al., 2017), the authors of the present manuscript claim that their data do show that SDC1 ablation does not impact on the generation of antibody-secreting plasma cells. The difficulty in enumerating plasma cells deficient for SDC1 is that SDC1 is a key cytometric marker for plasma cells. In particular here single cell transcriptome signatures would have been extremely helpful. In their absence, the authors base their classification on surrogate markers like CD19 and size (EVFig. 4) with erratic gate settings which are not defining separate populations (lymphocyte, FSA and CD19 gates). While in conjunction with CD138, these gates may define plasma blasts or even plasma cells, in the absence of CD138, i.e. in the SDC1-deficient B lineage cells, they obviously do not. There is considerable spillover from the adjacent populations and the true numbers and differentiation stages of antibody-secreting B lineage cells remain enigmatic. Even more so, the authors also show the numbers of antigen-binding "plasma cells" in Figs. 3F and I, which at least for Ig-secreting cells means that these cells are not yet fully mature plasma cells, which would not express membrane-bound antibodies, but rather either B cell blasts or plasma blasts, thus weakening the claim that SDC1 does not impact on plasma cell differentiation even further.

The authors then analyse the regulation of SDC expression in activated B cells by the costimulators IL21, IL4 and CD40L. Here they claim that SDC1 is upregulated and SDC4 downregulated in antigen-activated B cells, and IL21 enhances, while IL4 and CD40L inhibit these SDC regulations. What is not clear from the data shown is to what extent IL21, IL4 and antiCD40 impact on the differentiation of activated B cells into plasma cells versus their proliferation as B cells, i.e. whether their effect is directly on SDC expression or indirectly on B cell activation (Fig. 4).

Finally, in Figs 5 and 6, the authors use heparin (Fig. 5) and heparinase (Fig. 6) to demonstrate that the GAG heparin sulfate can (down)regulate B cell activation. These data are not directly linked to SDC1 or 4, and thus remain a bit preliminary. As are the data on S6 phosphorylation. As a net result, the molecular mechanism of SDC4 attenuation of antigen-mediated activation of B cells remains to be elucidated.

Referee #1:

McKenzie et al describe "The role of syndecans in B cell development, activation and plasma cell formation".

Plasma cells, by secreting antibodies, are obviously central to humoral immunity. Plasma cells are characterized by high expression of CD138, but the role of this glycosaminoglycan-containing protein besides serving as a marker of plasma cells for immunologists is incompletely understood.

Here, the authors study the expression and function of CD138 (Syndecan-1) and of another syndecan expressed in B cells (syndecan-4) using state of the art techniques in at least 10 B cell stages between pre-pro B cells and plasma cells. It is noteworthy that their study stopped at plasma cell generation and was not designed to study how PCs are maintained in the long-term. The work is divided in 4 main blocks.

In a first part, the detail surface expression of CD138 and Syndecan4 is studied. In a second part, authors monitor, using bone marrow chimera, how stem cells knock-out for Sdc1 (CD138) or Sdc4 compare to WT to reconstitute B cell populations. Results show that neither Sdc1 nor Sdc4 are essential for B cell development. Indeed plasma cells can be produced and are alive in the absence of one or the other. In both ko, B cell development is impaired about 50% usually at the stage where these molecules are most highly expressed. This indicates a positive but non-essential function of Sdc1 and Sdc4 during development.

In a third part, regulation of SDC1 and SDC4 expression was studied in naïve B cells that express SDC4 but not SDC1. Direct stimulation of the BCR permanently decreased SDC4 expression and transiently increased SDC1 expression. Other stimuli tested (IL4, IL21, anti-CD40 [all produced by or mimicking T helper cells]) had no effect alone, but tended to maintain the status quo of SDC1 and 4 expression upon BCR stimulation (and therefore go against the BCR effect on SDC1 and 4, but without preventing proliferation). The exception was IL21 that further promoted transient BCR-mediated CD138 expression. Consistent results of SDC4 downregulation and SDC1 expression were observed when B cells were stimulated with antigen and T helper cells in vivo.

In the fourth part, authors studied the effect of adding moderate or excess amounts of soluble glycosaminoglycans to naïve B cells {plus minus} cognate antigen. In a "reverse" experiment, they digested glycosaminoglycans on B cells with heparinase. Although effects were present and sometimes spectacular (addition of glycosaminoglycans increased B cell number up to almost 10-fold; addition of glycosaminoglycans reduced activation markers CD69 or Fas or loss of SDC4 induced by BCR stimulation; removal of cell-bound glycosaminoglycans with heparitinase increases CD69 expression post BCR stimulation), I found it difficult to understand how authors conclude that glycosaminoglycans and in particular SDC4 have an inhibitory function on B cells, and to figure out by which mechanism this could be.

We conclude that SDC4 and glycosaminoclycans (GAG) limit B cell activation based on three independent lines of evidence:

1. GAG reduce CD69 upregulation, activation-induced cell death (Fig. 6A) and FAS expression (Fig. 4E) in response to antigen. This indicates a role for GAG in BCR signalling, which is in line with the potent modulation of Syndecan expression by antigen encounter (Figure 3).
2. GAG removal by heparinase resulted in increased activation (CD69 upregulation) in response to low-affinity (HEL3x) antigen stimulation (Fig. 5B)
3. SDC4 deletion increased baseline S6 and Syk phosphorylation and antigen-induced S6 phosphorylation (Fig. 5C and new Fig. EV6H).

While our experiments do not directly link SDC-linked GAG to these effects, an aspect we highlight in the discussion, we think that these data convincingly support the novel conclusion that GAG and SDCs limit BCR signalling.

In summary, the experiments are very well performed, figures are of high quality, the introduction is to the point and some important conclusions are fully supported by data (SDC4 and CD138 are not essential for B cell development and production of plasma cells). The conclusion that syndecans (at least SCD4) are inhibitory for naïve B cells is probably supported by data, but this could be better explained. Often, conclusions are written in vague (or cautious) terms that do not help the reader understand what is going on (eg title: "The role of syndecans in B cell development, activation and plasma cell formation." OK, but what is this role? Eg abstract: "Collectively, our results reveal SDCs as regulators of the B cell response to antigen, with GAGs modulating B cell activation and survival." The terms "regulator" and "modulating" gives no indication on the direction or amplitude of the modulation. Etc...

We thank the reviewer for their positive assessment of our work and the suggested improvements. We have changed the title and re-phrased the abstract and conclusions to better reflect the findings of our study.

Major point 1: The authors should clarify their descriptions of Fig 5 and Fig 6 investigating the role of GAGs in

naïve B cell activation. For example, Fig 5A shows that heparin has no effect on the number of naïve B cells in the absence of antigen stimulation, but that in the presence of the antigen (HEL), there are about 10-fold more B cells in the presence of heparin than in its absence. It is very difficult for the reviewer to reconcile this result with the final conclusion of the authors that GAGs have an inhibitory role on B cells. One possibility is that under these conditions, the high affinity ligand induces death of the stimulated cells (AICD) which would explain the strongly reduced cell number of the condition "no GAGs" in the presence of antigen compared to the absence of antigen. AICD could be due to upregulation of FAS which is counteracted to some extent by the presence of heparin. As FAS is an activation marker, then heparin would negatively regulate activation, keep cells under the threshold of AICD and prevent cells from dying. If this is the case, it could be explained better, eg in a wrap up sentence. But then it is unclear why in Fig 5C that tests the effect of short-term naïve B cells exposure to heparin there is no difference in B cell number in the "No GAG" condition minus or plus antigen, and that this is also not seen in Fig 5F in No GAG minus or plus antigen. Fig 6A however resembles Fig 5A in that naïve B cell number is drastically reduced upon stimulation with antigen (probably by AICD). This defect is attenuated by heparin, but never to the extent seen in Fig 5A. Experimental variability? Different experimental conditions?

In the absence of antigen-induced activation, SW_{HEL} B cells die over the course of 3 days and the extent of this death slightly varied between experiments, something we can't explain as yet. However, the key point we make in these experiments is that a dose of heparin equal to that used for therapeutic purposes did not effect the survival/expansion of cells in the absence of antigen but greatly increased B cell numbers in the presence of antigen. Mechanistically, this is due to the reduction in AICD, an instant effect of high-affinity antigen encounter. We discuss AICD later in the manuscript, in conjunction with Figure 5A (formerly 6A). We agree it is important to highlight the experimental variability and have added the following sentence:

"Of note, the survival of SW_{HEL} B cells in the absence of antigen varied between experiments (Fig. 4A, C) but the effect of heparin was highly consistent across experiments."

Also, it would be useful for the reader if authors could integrate results of Fig 4 with their conclusion that syndecans have an inhibitory role on B cells. For example, is it conceivable that T cell help would prevent premature differentiation of cycling B cells in antibody secreting cells and plasma cells by among others, preventing CD138 expression and preventing SDC4 down-regulation, with the aim of achieving a better cell amplification before terminal differentiation?

We have added the following sentence to the discussion:

"T cell help via cytokines or CD40 signaling stimulated or retained SDC4 expression over consecutive cell divisions following naïve B cell activation and it is conceivable that this mechanism facilitates pre-GC B cell expansion."

Minor point 2: In the experiment of Figure 4, naïve B cells not only express anti-HEL BCR, but also come from a Rag1-ko background. It would be helpful in all figures using B cells with anti-HEL BCR to mention if they are also Rag1-ko or not.

We have adjusted the Figure legends accordingly as all SW_{HEL} B cells used in this study were RAG1 deficient.

Minor point 3: In Fig 4A, the CTV profile indicates that CD138 that is not expressed in naïve B cells was induced by HEL even before division, and then fully lost again after 3 divisions. This profile looks exactly like the one of Fig 4B with SDC4, except that here SDC4 was already present and is just lost upon HEL stimulation within 3 divisions. This could be better explained in the text. What justifies this very transient expression is a mystery for me. It cannot be directly linked to a differentiation to PC. Are Fig 4A,B comparable in terms of cells and HEL activation to Fig 5 and 6? If yes and if CD69 and/or FAS staining were included in these experiments (of Fig 4A,B), and if SDC4 indeed prevents activation, one would expect to see CD69 go up as cells divide in the Media only + HEL condition, but not in the anti-CD40 + HEL condition. If this data is available, please show the results (and possibly revise the conclusion of the inhibitory role of SDCs as necessary). If these stainings were not included, I let authors decide whether they would like to do it, but I do not insist that the experiment should be done.

We did not include FAS and CD69 in this experiment and have added 2 sentences to better describe the data:

"As expected, naïve SW_{HEL} B cells lacked CD138 but expressed high levels of SDC4."

...

"PC express high levels of CD138 but the transient CD138 expression induced by antigen stimulation is not associated with the initiation of the PC program, which only starts after > 5 cell divisions (Dvorscek et al, 2022; Hasbold et al, 2004; Scharer et al, 2018)."

Minor point 4: It is unclear how addition of heparin, or digestion with heparinase would interfere with SDC4 to

exert its effect on B cells. A) GAGs of SDC4 inhibit B cell activation directly (e.g. by binding the BCR). Adding heparin allows to inhibit B cells even more (e.g. by binding more BCR). The digestion with heparinase removes GAGs and less partner is inhibited. B) GAGs of SDC4 bind a positively charged partner (APRIL, IL6, IL21?) that is displaced by an excess of heparin and thus stops activating B cells. What happens if naïve B cells are digested with heparinase, then washed, then stimulated with heparin? If heparin can bind and inhibit the same target as SDC4, this should cancel the action of heparitinase (I do not insist to have this experiment done).

We thank the reviewer for these interesting thoughts but consider it very unlikely that soluble heparin only acts on molecules that also interact with SDC4-bound proteoglycans, in particular given the comparatively strong effect of heparin.

Referee #2:

Syndecans can play important roles as modulators of cytokine and growth factor responses and can mediate attachment of cells. They can also act as co-receptors for GPRs. This report aims at addressing the abundance and function of two Syndecans, Sdc1 and 4, in B cells. To this end, bone marrow of Cas9tg mice was infected with lentiviral vectors encoding specific sgRNAs and a BFP reporter. Infected BM along with RAG1^{-/-} BM was transplanted into irradiated WT mice and B cell development and activation were analyzed. While CD138 peaks at pre B cells and in plasma cells, Sdc4 becomes up-regulated in pre and immature B cells and down-regulated upon B cell activation as well as in memory B cells, however, its expression differs among memory B cell subsets.

Functionally, deletion of both CD138 and Sdc4 disfavoured early B cell development. While Sdc4 B cells reached control frequency, the disadvantage of CD138 targeted B cells remained until the immature B cell stage. Immunization of recipient mice with NP-KLH did not reveal any effects of CD138 or Sdc4 deletion on the frequency of deleted germinal center B cells or plasma cells.

As to B cell activation, the SwHEL system was used in combination with IL-4, IL-21 and anti CD40. While IL-21 promoted HEL induced CD138 upregulation, IL-4 and CD40 suppressed it. The reverse was observed for Sdc4.

To address functional effects by putative Sdc ligands such as Polybrene and Heparin, activated B cells were co-cultured with either substance or treated with Heparinase. Heparin acted as B cell mitogen and is discussed as putative TLR4 ligand, hence being not so much relevant for human B cells in that respect. On the other hand, Heparin (HEL stimulation) suppressed AICD and CD69 expression. Heparinase (3xHEL) had no effect on AICD but increased CD69 expression. Sdc4 limited pS6 in resting and anti Ig activated B cells.

In summary, the presented data serve as a good starting point to pinpoint the role of CD138 and Sdc4 in B cells.

1. There are only few functional data regarding CD138 or Sdc4 knock-out, but both appear to affect early B cell development, which is interesting. What is the mechanism behind the disadvantage of CD138 and Sdc4 deficient pre/pro/pre B cells? pS6 needs to be downregulated to ensure proper small pre B cell development and Sdc4 appears to modulate pS6 in activated B cells. Given that the authors outstanding expertise in BM and stem cell cultures, would it not be feasible to set up stem cell / IL-7 cultures with CD138 or Sdc4 targeted stem cells and analyze pre B cell differentiation *in vitro*, defining proliferation/apoptosis/signaling/small pre B cell differentiation (using CD19/B220/CD2/IgM staining)? Pre BCR ligands should be discussed. It is also possible that *in vitro* cultures show no effect as the potential Sdc ligand may only be present in the bone marrow. However, I believe that the role of Sdcs in B cell development should be explored more here.

We thank the reviewer for this excellent suggestion. We have now performed *in vitro* B cell differentiation assays using bone marrow from *Sdc1* or *Sdc4* targeted mice. These cultures solely depend on the presence of IL-7 and resulted in a strong (10-fold) expansion of B cell precursors (new Figure EV4C), which was associated with a loss of pro-B cells and an accumulation of large and small pre-B cells (new Figure EV4D). As shown in new Figures EV4E-F, the effects of *Sdc1* and *Sdc4* deletion closely resembled *in vivo* results, suggesting a role for SDCs in IL-7 signalling. This is also in line with studies by Paul Kincade and colleagues (PMID 9864155) who showed a key role of heparan sulfate proteoglycans in IL-7-dependent B cell development.

2. Was the pS6 in activated B cells normalized to total S6? Can pS6 be blocked by Rapamycin? I think the conclusion that Sdc4 is a negative regulator of BCR signaling is a bit premature. Is BCR induced pSyk/PLCg2/Ca²⁺ affected or are the observed effects downstream of proximal BCR signaling?

pS6 was not normalized to total S6. As shown in NEW Figure EV6G, S6 phosphorylation can be blocked by rapamycin treatment. To assess BCR signalling, we sorted BFP+ and BFP- cells from *Sdc1* and *Sdc4* targeted mice and analyzed Syk phosphorylation by phosphoflow. As shown in NEW Figure EV6F, baseline Syk

phosphorylation was increased in *Sdc4* but not *Sdc1* targeted mice while pSyk following BCR cross-linking was similar. The latter could be due to the high affinity of BCR cross-linking antibodies.

3. Conceptually, it would be important to assess the effects of Heparin and Heparinase also on *Sdc4*KO B cells.

The effect of heparin was much stronger than the deletion of *Sdc4*, making it highly likely that heparin treatment will not additionally affect *Sdc4*ko cells. In keeping with this, the presence of heparin reduced Syk phosphorylation irrespective of SDC4 (Reviewer Figure 1A). We also analysed the effect of heparinase treatment on Syk phosphorylation, with the data showing no difference between non-targeted and *Sdc4* targeted cells (Reviewer Figure 1B). A detailed investigation of the effect of heparinase on antigen-mediated activation of SDC4 deficient cells would require the generation of SDC4 deficient SW_{HEL} mice to stimulate B cells with antigens of various affinity. This is associated with several months of significant mouse breeding efforts (generation of Cas9 transgenic SW_{HEL} mice) and goes beyond what can be done within the timeframe for manuscript revision. We did not include the results shown in Reviewer Figure 1 in the manuscript as they add little additional information.

*Reviewer Figure 1. (A) Syk phosphorylation in the presence or absence of heparin comparing non-targeted (BFP-) to *Sdc4* targeted (BFP+) cells. (B) Syk phosphorylation in heparinase treated non-targeted (BFP-) or *Sdc4* targeted (BFP+) B cells. Colours indicate cells from individual mice, assayed in duplicates.*

4. What happens if B cells are incubated with both Polybrene and Heparin?

We have now performed an experiment in which we incubated B cells with antigen, heparin and polybrene. As shown in new Figure EV6F, polybrene reversed the effect of heparin in a dose-dependent manner.

5. Why did the authors use HEL in Figure 6A and 3 x HEL in Figure 6B? This is a bit misleading and hard to grasp.

HEL has high affinity for the SW_{HEL} BCR and maximally activates B cells making the use of HEL as an antigen not ideal to study molecules that promote BCR signalling. We therefore used the lower affinity HEL3x antigen to investigate if heparinase treatment promoted antigen-mediated activation.

Referee #3:

The manuscript of McKenzie and colleagues provides a comprehensive overview on expression of the Glycosaminoglycan (GAG) containing cell surface proteins SDC1 and SDC4 during differentiation and activation of B lymphocytes. Based on this overview, the impact of genetic ablation of SDC1 or SDC4 in mice, and the impact of soluble GAGs on human and murine B cell activation is analysed, with the key claims that SDC4 is a negative modulator of B cell activation and SDC1 (CD138) is not relevant for differentiation of activated B cells into plasma cells. As such, the manuscript adds important information on the regulation of B cell differentiation and activation by GAGs, of interest to the readership of EMBO Reports, in light of the central role of B cells in providing humoral immunity and autoimmunity. However, some issues have to be clarified upfront. The major drawback seems to be that the entire readout of expression is based on cytometric data, and not adjusted to (single cell) transcriptome data, for staging of B cell differentiation. This would clearly put the manuscript on a different level.

The descriptive part of the manuscript, i.e. Figs 1 and 2, and EVFigs 1 and 2, is rather clear, as far as the expression pattern of SDC1 and SDC4 in the general steps of B cell differentiation is concerned. However, when

analysing the bimodal expression of SDC1 and 4 in experienced B cells, and using CD62L and CD44 to discriminate between recent GC emigrant MBC (CD62L+CD44-) and established MBC (CD62L+CD44+), the data shown in EVFig 2C show that the mice analysed did not have any CD62L+CD44+ MBC, for whatever reason, maybe the mice were just too young and immunologically inexperienced. However, in Fig. 2A bimodal distributions are shown for CD62L-CD44- MBC and CD44-CD62L+ MBC, not for CD44+CD62L+, but then in Fig. 2B, frequencies are provided for these MBC. These data are not convincing. The differential expression of SDC1 and 4 on MBC remains enigmatic, the data provided are inconsistent, and mice with significant numbers of advanced MBC should be analysed.

We have now performed experiments to study defined, antigen-specific MBC. As such, we immunized mice with the model antigen NP-KLH and analyzed MBC three weeks thereafter. We identified MBC subsets based on CD62L/CD44 expression and on a more common approach using PD-L2 and CD80. NP-binding MBC were clearly identified and showed an expected subset distribution (new Fig. EV2). All MBC subsets showed low, heterogeneous SDC4 expression, with around twice as many PD-L2⁺CD80⁻ MBC expressing SDC4 compared to all other subsets.

When it comes to the controversial point, and as the authors claim key statement, of whether SDC1 (CD138) does impact on plasma cell differentiation, as suggested by an earlier publication (McCarron et al., 2017), the authors of the present manuscript claim that their data do show that SDC1 ablation does not impact on the generation of antibody-secreting plasma cells. The difficulty in enumerating plasma cells deficient for SDC1 is that SDC1 is a key cytometric marker for plasma cells. In particular here single cell transcriptome signatures would have been extremely helpful. In their absence, the authors base their classification on surrogate markers like CD19 and size (EVFig. 4) with erratic gate settings which are not defining separate populations (lymphocyte, FSA and CD19 gates). While in conjunction with CD138, these gates may define plasma blasts or even plasma cells, in the absence of CD138, i.e. in the SDC1-deficient B lineage cells, they obviously do not. There is considerable spillover from the adjacent populations and the true numbers and differentiation stages of antibody-secreting B lineage cells remain enigmatic. Even more so, the authors also show the numbers of antigen-binding "plasma cells" in Figs. 3F and I, which at least for Ig-secreting cells means that these cells are not yet fully mature plasma cells, which would not express membrane-bound antibodies, but rather either B cell blasts or plasma blasts, thus weakening the claim that SDC1 does not impact on plasma cell differentiation even further.

To confirm that our gating strategy indeed identifies PC, we have applied that approach to prdm1-tdTomato reporter mice. In these mice all PC are identified by tdTomato expression and >90 % of cells identified as PC by our gating strategy expressed tdTomato (new Fig. EV4B). This provides clear evidence that our approach is applicable for PC analysis.

Regarding the analysis of NP-specific PC, we would like to note that most early PC retain some surface BCR expression, and NP-specific PC can be readily detected (please see for example Fig1A from PMID 27924814 or Fig. 2A from PMID 22529295). We agree that the NP-specific PC are unlikely to be mature, long-lived PC given the analysis was done 7 days post immunization and PC maturation occurs gradually over time. To highlight this important point, we have now included the following sentences on page 5 of the revised manuscript:

"Similarly, Sdc1 targeting did not influence the representation of these cells among splenic GC and PC compared to naïve B cells when total (Fig. 2E) or NP-binding GC B cells and PC (surface immunoglobulin expression by early PC allows detection of antigen-specific cells (Blanc et al, 2016; Bortnick et al, 2012)) were compared (Fig. 2F), again arguing against a role of CD138 for B cell activation or PC differentiation. It is important to note that our data do not allow conclusions about the potential role of CD138 in regulating PC lifespan as the total PC pool is constantly replenished and long-lived PC make up only a small fraction of all PC (Robinson et al., 2023; Robinson et al., 2022). However, at the level of analysis conducted here, and aside from a possible drop in small pre-B cells, CD138 had no or only a minor role in B cell development and was dispensable for antigen-induced activation and PC differentiation."

The authors then analyse the regulation of SDC expression in activated B cells by the costimulators IL21, IL4 and CD40L. Here they claim that SDC1 is upregulated and SDC4 downregulated in antigen-activated B cells, and IL21 enhances, while IL4 and CD40L inhibit these SDC regulations. What is not clear from the data shown is to what extent IL21, IL4 and antiCD40 impact on the differentiation of activated B cells into plasma cells versus their proliferation as B cells, i.e. whether their effect is directly on SDC expression or indirectly on B cell activation (Fig. 4).

Plasma cell differentiation has yet to occur at the time point analyzed. We have now added the following sentence to clarify that:

“PC express high levels of CD138 but the transient CD138 expression induced by antigen stimulation is not associated with the initiation of the PC program, which only starts after > 5 cell divisions (Dvorscek et al, 2022; Hasbold et al, 2004; Scharer et al, 2018).”

Finally, in Figs 5 and 6, the authors use heparin (Fig. 5) and heparinase (Fig. 6) to demonstrate that the GAG heparin sulfate can (down)regulate B cell activation. These data are not directly linked to SDC1 or 4, and thus remain a bit preliminary. As are the data on S6 phosphorylation. As a net result, the molecular mechanism of SDC4 attenuation of antigen-mediated activation of B cells remains to be elucidated.

Analyzing the detailed molecular mechanism of how SDC4 regulates B cell activation goes beyond the scope of this study. We do think, however, that our data add valuable information about the role of SDCs in B cell biology and the effects of soluble proteoglycans provide an important advance in our understanding how B cell activation can be regulated.

Dear Dr. Quast,

Thank you for the submission of your revised manuscript to our editorial offices. I have now received the report from the three referees that were asked to re-evaluate the study, you will find below. As you will see, the referees now fully support the publication of the study in EMBO reports. Please correct the typo mentioned by referee #2.

Before I can proceed with formal acceptance, I have these editorial requests I ask you to address in a final revised manuscript:

- Please provide the abstract written in present tense.

- Please reduce the number of keywords to 6 and order the manuscript sections like this, using these names:

Title page - Abstract - Keywords - Introduction - Results - Discussion - Methods - Data availability section - Acknowledgements (including the funding information) - Disclosure and Competing Interests Statement - References - Figure legends - Expanded View Figure legends

- Please upload individual production quality figure files as .eps, .tif, .jpg (one file per figure), also for the EV figures. Please upload these as separate, individual files upon re-submission. Moreover, please use the nomenclature Figure EVx for these in their legends and check again all the callouts.

- Please make sure that all the funding information is also entered into the online submission system and that it is complete and similar to the one in the acknowledgement section of the manuscript text file. Presently, grants 'Monash University Research Training Program stipend and Career Advancement Award provided by the Australian and New Zealand Society for Immunology' and 'Monash University Future Leader Fellowship' are missing from the submission system. Please check.

- The Data availability section (DAS) is restricted for information on primary datasets produced in a study (e.g. RNA-seq, ChIP-seq, structural and array data) that are deposited in a public database. If no primary datasets have been deposited, please state this here (e.g. 'No primary datasets have been generated and deposited'). Please remove all further text not related to externally deposited datasets from this section.

- Please check again that the number "n" for how many independent experiments were performed, their nature (biological versus technical replicates), the bars and error bars (e.g. SEM, SD) and the test used to calculate p-values is indicated in the respective figure legends. Please also check that all the p-values are explained in the legend, and that these fit to those shown in the figure. Please provide statistical testing where applicable. Please avoid the phrase 'independent experiment', but clearly state if these were biological or technical replicates. Please also indicate (e.g. with n.s.) if testing was performed, but the differences are not significant. In case n=2, please show the data as separate datapoints without error bars and statistics. See also:

<http://www.embopress.org/page/journal/14693178/authorguide#statisticalanalysis>

If n<5, please show single datapoints for diagrams. Presently, several diagrams have no or only partial statistics or 'n.s.' is missing (e.g. please check panels 1D, 1F, 2D-I, EV4D-F, EV5C and EV6B-I). Moreover:

- Please define the annotated p values ****/**/*/* as well as provide the exact p-values for the same in the legend of figure 5A, EV4 C as appropriate.

- Please note that the exact p values are not provided in the legends of figures 1F, 2B, C; 3C, D, 4A-F; 5A-C; EV5 B.

- Please note that information related to n is missing in the legends of figures 3C, D; 4F, EV5 C, EV6 B, C, D, E, F, G.

- Please note that the error bars are not defined in the legends of figures 1F, 4B, D; EV5 C, EV6 B, C, D, E, F.

- There seems to be a re-use of panels between Figure EV2A (first row, 'All events' and 'Live') and Figure EV5D (first row, 'All events' and 'Live'). If this is intended, please clearly explain and indicate this in the respective figure legends.

In addition, I would need from you uploaded separately:

- a short, two-sentence summary of the manuscript (not more than 35 words).

- two to four short (!) bullet points highlighting the key findings of your study (two lines each).

- a schematic summary figure as separate file that provides a sketch of the major findings (not a data image) in jpeg or tiff format (with the exact width of 550 pixels and a height of not more than 400 pixels) that can be used as a visual synopsis on our website.

Achim Breiling
Editor
EMBO Reports

Referee #1:

I thank authors for their answers and I have no further comments

Referee #2:

I thank the authors for addressing the comments adequately.

I found one typo: l 144, PLIMP-1.

Referee #3:

The authors have responded perfectly to my previous concerns and modified the manuscript accordingly, added new data, in particular on MBC, to clarify previous inconsistencies. From my point of view, the manuscript should now be accepted.

Dear EMBO Reports team, dear Achim,

Please find below a point-by-point-response to all requests.

- Please provide the abstract written in present tense.
The abstract is now written in present tense.

- Please reduce the number of keywords to 6 and order the manuscript sections like this, using these names: Title page - Abstract - Keywords - Introduction - Results - Discussion - Methods - Data availability section - Acknowledgements (including the funding information) - Disclosure and Competing Interests Statement - References - Figure legends - Expanded View Figure legends
Keywords were reduced to 5. Section order has been changed accordingly.

- Please upload individual production quality figure files as .eps, .tif, .jpg (one file per figure), also for the EV figures. Please upload these as separate, individual files upon re-submission. Moreover, please use the nomenclature Figure EVx for these in their legends and check again all the callouts.
Nomenclature changed accordingly, individual HQ figures uploaded and callouts re-checked.

- Please make sure that all the funding information is also entered into the online submission system and that it is complete and similar to the one in the acknowledgement section of the manuscript text file. Presently, grants 'Monash University Research Training Program stipend and Career Advancement Award provided by the Australian and New Zealand Society for Immunology' and 'Monash University Future Leader Fellowship' are missing from the submission system. Please check.
Checked and added to the online system.

- The Data availability section (DAS) is restricted for information on primary datasets produced in a study (e.g. RNA-seq, CHIP-seq, structural and array data) that are deposited in a public database. If no primary datasets have been deposited, please state this here (e.g. 'No primary datasets have been generated and deposited'). Please remove all further text not related to externally deposited datasets from this section.
Changed accordingly.

- Please check again that the number "n" for how many independent experiments were performed, their nature (biological versus technical replicates), the bars and error bars (e.g. SEM, SD) and the test used to calculate p-values is indicated in the respective figure legends. Please also check that all the p-values are explained in the legend, and that these fit to those shown in the figure. Please provide statistical testing where applicable. Please avoid the phrase 'independent experiment', but clearly state if these were biological or technical replicates. Please also indicate (e.g. with n.s.) if testing was performed, but the differences are not significant. In case n=2, please show the data as separate datapoints without error bars and statistics. See also: <http://www.embopress.org/page/journal/14693178/authorguide#statisticalanalysis>
Nature and amount (n) of replicates clearly stated.

If n<5, please show single datapoints for diagrams. Presently, several diagrams have no or only partial statistics or 'n.s.' is missing (e.g. please check panels 1D, 1F, 2D-I, EV4D-F, EV5C and EV6B-I). Moreover:

- Please define the annotated p values ****/**/* as well as provide the exact p-values for the same in the legend of figure 5A, EV4 C as appropriate.

We now show all exact p values in the figures, and I have removed all 'stars'. I have added additional statistical comparisons but for some data, we have chosen not to perform statistical analysis for the following reasons:

- 1 B and D show Syndecan expression levels of on cell types and I do not think statistical analysis is relevant but can of course be provided if that is preferred.
- Figure 1F, statistical analysis of relevant comparisons provided.
- Figure 2D-I and EV4E, F show the representation of the cell population over consecutive differentiation stages and the key point is the trend over time rather than comparison within individual differentiation stages.
- EV5B,C: The most relevant statistical analysis would be comparing panels B and C which would require changing the figure outline. If requested, I could prepare a separate graph showing 'division number 0' only data and perform statistical analysis but I do not think this adds information.
- EV6B-E: Statistical analysis provided.
- EV6F: This is supporting data for Figure 4F and shows that the effect of polybrene is titratable. To analyze this statistically, I would have to do a correlation analysis which is difficult given the data is multi-dimensional (division number, cell-count and polybrene dose). I think the correlation between increased polybrene and reduced cell count is very clear and does not require statistical analysis.
- EV6G: There are not enough data points for statistical analysis. This data was provided in response to a reviewer request and is simply confirmatory, showing the well-known fact that rapamycin inhibits S6 phosphorylation.

- Please note that the exact p values are not provided in the legends of figures 1F, 2B, C; 3C, D, 4A-F; 5A-C; EV5 B.

Exact p values are now shown for all statistical analysis (all * removed).

- Please note that information related to n is missing in the legends of figures 3C, D; 4F, EV5 C, EV6 B, C, D, E, F, G.

Information added.

- Please note that the error bars are not defined in the legends of figures 1F, 4B, D; EV5 C, EV6 B, C, D, E, F.

Error bars defined (there are no error bars in 1F)

- There seems to be a re-use of panels between Figure EV2A (first row, 'All events' and 'Live') and Figure EV5D (first row, 'All events' and 'Live'). If this is intended, please clearly explain and indicate this in the respective figure legends.

Thank you for highlighting this, we have prepared the gating strategy figures on the same day and must have accidentally forgotten to exchange this panel because FSC/SSC look very similar for every sample. We have now provided the correct exemplary image for Figure EV5D.

In addition, I would need from you uploaded separately:

- a short, two-sentence summary of the manuscript (not more than 35 words).

- two to four short (!) bullet points highlighting the key findings of your study (two lines each).

- a schematic summary figure as separate file that provides a sketch of the major findings (not a data image) in jpeg or tiff format (with the exact width of 550 pixels and a height of not more than 400 pixels) that can be used as a visual synopsis on our website.

Provided.

Achim Breiling
Editor
EMBO Reports

Referee #1:

I thank authors for their answers and I have no further comments

Referee #2:

I thank the authors for addressing the comments adequately.

I found one typo: l 144, PLIMP-1.

The typo has been corrected.

Referee #3:

The authors have responded perfectly to my previous concerns and modified the manuscript accordingly, added new data, in particular on MBC, to clarify previous inconsistencies. From my point of view, the manuscript should now be accepted.

We thank the reviewers for their valuable contributions to improving our study.

Dr. Isaak Quast
Monash University
Immunology and Pathology
89 Commercial Road
Melbourne, Victoria 3004
Australia

Dear Dr. Quast,

I am very pleased to accept your manuscript for publication in the next available issue of EMBO reports. Thank you for your contribution to our journal.

Yours sincerely,
